# Snow conditions in northern Europe: the dynamics of interannual variability versus projected long-term change

Jouni Räisänen[1]

[1] Institute for Atmospheric and Earth System Research / Physics, University of Helsinki, P.O. Box 64, FI-00014 University of Helsinki, Finland

*Correspondence to*: Jouni Räisänen (jouni.raisanen@helsinki.fi)

**Abstract.** Simulations by the EURO-CORDEX regional climate models indicate a widespread future decrease in snow water equivalent (SWE) in northern Europe. This concurs with the negative interannual correlation between SWE and winter temperature in the southern parts of the domain, but not with the positive correlation observed further north and over the Scandinavian mountains. To better understand these similarities and differences, interannual variations and projected future changes in SWE are attributed to anomalies / changes in three factors: total precipitation, the snowfall fraction of precipitation, and the fraction of accumulated snowfall that remains on ground (snow-on-ground fraction). In areas with relatively mild winter climate, the latter two terms govern both the long-term change and interannual variability, resulting in less snow with higher temperatures. In colder areas, however, interannual SWE variability is dominated by variations in total precipitation. Since total precipitation is positively correlated with temperature, more snow tends to accumulate in milder winters. Still, even in these areas, SWE is projected to decrease in the future due to reduced snowfall and snow-on-ground fractions in response to higher temperatures. Although winter total precipitation is projected to increase, its increase is smaller than would be expected from the interannual co-variation of temperature and precipitation, and therefore insufficient to compensate the lower snowfall and snow-on-ground fractions. Furthermore, interannual SWE variability in northern Europe in the simulated warmer future climate is increasingly governed by variations in the snowfall and snow-on-ground fractions, and less by variations in total precipitation.

## 1 Introduction

Due to its location near the western margin of the Eurasian continent, northern Europe experiences large interannual variations in winter climate associated with variations in the atmospheric circulation (Tuomenvirta et al., 2000; Hansen-Bauer and Førland, 2000; Chen, 2000; Lehtonen, 2015; Saffioti et al., 2016; Räisänen, 2019). An example of a particularly anomalous winter was winter 2019/20, when strong westerly flow from the Atlantic Ocean and intense cyclone activity resulted in both unseasonally mild temperatures and very abundant precipitation (Figs. 1a-b). Record-breaking positive anomalies of 3-5 °C in the November-to-March mean temperature extended from southern Sweden to southern and central Finland, the Baltic States and western Russia, whereas the precipitation surplus was unusually large especially in Finland.

Yet the most remarkable feature of this winter were the dramatic regional contrasts in snow conditions, particularly in Finland (Fig. 1c). Based on the ERA5-Land data set (Muñoz Sabater, 2019), the March mean snow water equivalent (SWE) was record low since at least 1982 in southern Finland but record large in the north (as indicated by the stippling in Fig. 1c). This contrast was confirmed by station measurements of snow depth. In Helsinki at the south coast of Finland (H in Fig. 1),

there were only 9 days with a measurable (≥ 2 cm) snowpack and the largest snow depth was just 3 cm, compared with a previous all-time-low of 15 cm from year 1964. By contrast, Sodankylä in central Finnish Lapland (S in Fig. 1) reached a snow depth of 127 cm on 15 April 2020, exceeding its previous record of 119 cm from April 2000.

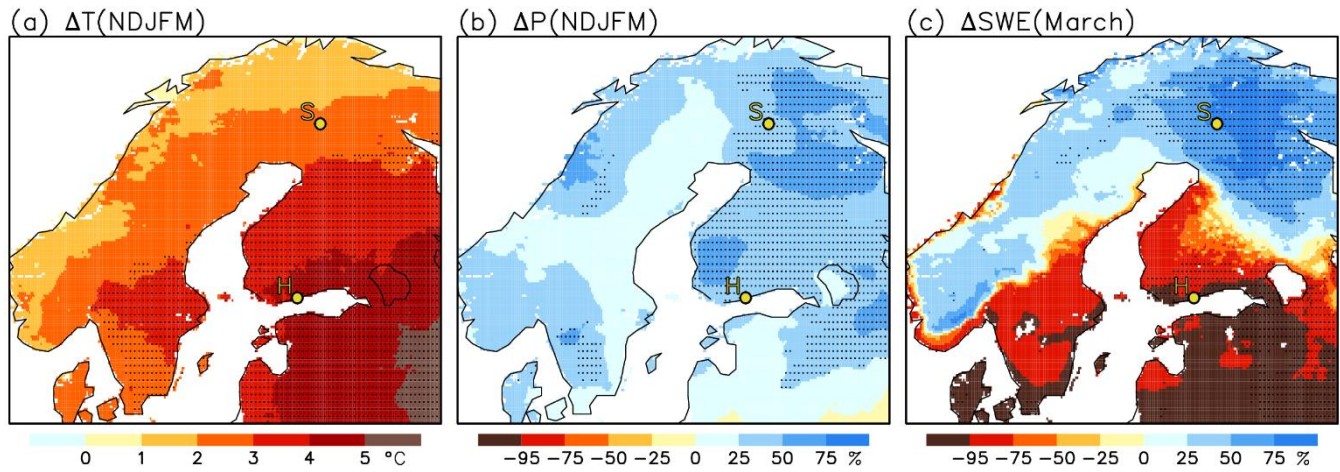

**Figure 1.** Anomalies of mean temperature (a) and precipitation (b) in November-March 2019/20, and the anomaly in SWE in March 2020
(c) compared with the corresponding mean values in the 39-winter period 1981/82-2019/20, based on ERA5-Land. Stippling indicates areas where the values in 2019/20 were either above or below the range in the previous 38 winters. The locations of Helsinki (H) and Sodankylä (S) are shown with yellow circles.

Although the SWE anomalies in winter 2019/20 were unusually large, their geographical distribution was typical for other mild winters. The correlation between the winter (defined here as November-to-March, NDJFM) mean temperature and

March mean SWE is negative in southern parts of northern Europe (e.g., southern Sweden, southern Finland, the Baltic states) as well as coastal Norway, but mostly positive further north and over the Scandinavian mountains (Fig. 2a). There are two main ingredients in this pattern: a generally positive correlation between winter temperature and precipitation (Fig. 2b), and geographical variations in the mean winter temperature (isotherms in Fig. 2). In the coldest areas in northern Europe (Lapland and Scandinavian mountains), nearly all precipitation falls as snow even in mild winters and melt episodes are

uncommon, enabling larger snow accumulation when both temperature and precipitation are above the average. In warmer regions, however, both the phase of precipitation and the occurrence of mid-winter melt events are much more sensitive to variations in temperature. As a result, the correlation between total winter precipitation and March mean SWE is negative in some of the milder regions, although it is strongly positive in the north and over the Scandinavian mountains (Fig. 2c).

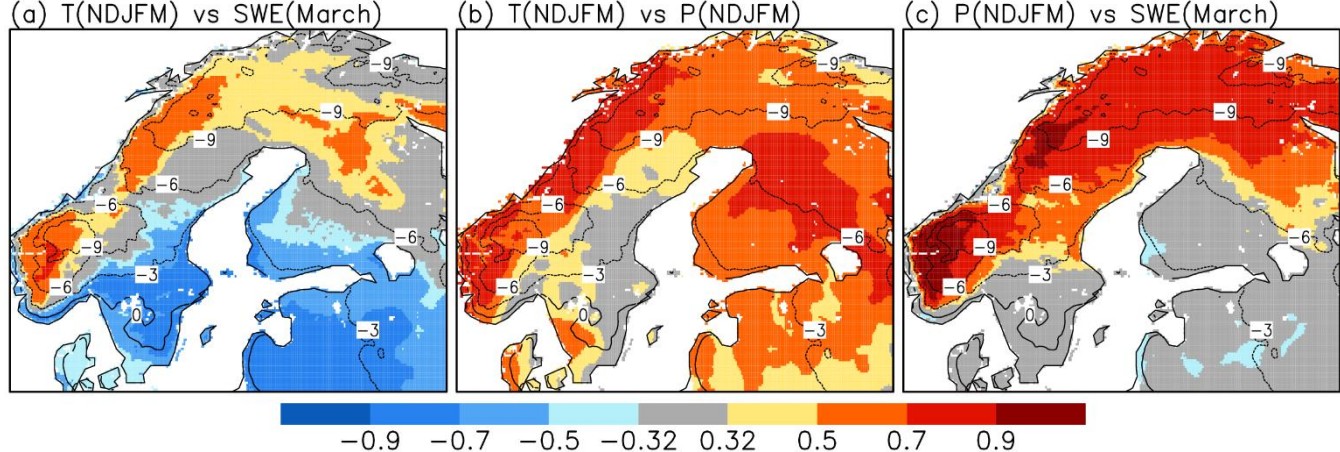

**Figure 2.** Correlation between (a) NDJFM mean temperature and March mean SWE, (b) NDJFM mean temperature and precipitation, and (c) NDJFM mean precipitation and March mean SWE in winters 1981/82 to 2019/20. Correlations not significant at 5% level ($|r| < 0.32$) are shown in grey. The contours show the 39-winter mean NDJFM temperature based on ERA5-land.

The mild temperatures that prevailed in winter 2019/20 made it tempting to consider this winter as an analogy of what would be experienced in a warmer future climate. For example, when lecturing a climate change course in University of Helsinki, I asked my students whether they believed that the snow conditions in this winter were a good analogy for the future. The majority answered positively, reasoning that (i) the higher temperatures would reduce the amount of snow in southern Finland but (ii) the impact of the warming would be overcompensated by increased precipitation in Lapland, where the winter mean temperature would still be well below zero even in the end of this century. Yet, climate model projections suggest that only the first half of this reasoning is correct. Both global and regional climate models tend to project a future decrease in snow amount in all of Finland (Fig. 13 in Räisänen and Eklund, 2012), although the decrease is smaller in northern than in southern Finland. Characteristics similar to the projected future SWE changes are also revealed by in-situ measurements of snow depth in Finland in years 1961-2014 (Luomaranta et al., 2019), although with a lower signal-to-noise ratio. These measurements show a decreasing trend in snow depth in most locations in Finland throughout the winter season, although less systematically in the north than in the south. Note, though, that snow depth is affected by snow density as well as SWE. For the longer period 1909-2008, Irannezhad et al. (2016) report a decrease in winter maximum SWE, as calculated by a temperature-index snowpack model from daily temperature and precipitation observations, at all their three study locations in Finland.

Motivated by these observations and model projections, this paper aims to elucidate and compare the dynamics of (i) interannual variability and (ii) projected future change in the amount of snow, particularly in Finland but also elsewhere in northern Europe. The analysis is built on a diagnostic method introduced by Räisänen (2008), which allows one to

decompose changes and anomalies in SWE to the contributions of three main factors: total precipitation, fraction of solid

precipitation (snowfall fraction), and the fraction of accumulated snowfall that has not yet melted and thus remains on ground (snow-on-ground fraction). Using this method, three main questions are explored: (i) which factors control the interannual variability of snow amount in northern Europe, (ii) how does the dynamics of the interannual variability differ from that of the projected long-term climate change, and (iii) how does the projected climate change affect the dynamics of interannual variability.


The significance of this research in a wider perspective is twofold. First, a better understanding of the processes involved in the interannual variability and long-term trends of snow conditions is valuable for model developers, helping to focus the development work towards the most important processes. For example, the findings in this paper suggest that, in areas with relatively mild winters like southern Finland, it is imperative to calculate snowmelt accurately for the realistic simulations of

both the interannual variability and future trends of snow amount. Second, the current results bear an important message for climate impact researchers and the general audience, by showing why the snow conditions in individual mild winters are not a perfect analogy for what to expect in the future.

In the following, the data sets used and the methods applied in analysing the data are first described (Sections 2-3). After

illustrating the decomposition method in Section 4, the main results of the data analysis are covered in three sections that address each of the questions (i)-(iii) listed above. Thus, section 5 focuses on interannual variability of SWE in the past four decades, Section 6 on future changes in long-term mean SWE simulated by regional climate models (RCMs), and Section 7 on the changes in interannual SWE variability that accompany the changes in mean climate. The conclusions are given in Section 8.

**2. Data sets**

**2.1 ERA5-Land**

The analysis in Sections 4-5 is based on ERA5-Land (Muñoz Sabater, 2019). ERA5-Land is a land-only rerun of the European Centre for Medium Range Weather Forecasts (ECMWF) ERA5 reanalysis (Hersbach et al., 2020), produced by forcing the H-TESSEL land surface model (Balsamo et al., 2009; Dutra et al., 2010) with ERA5 meteorological output

downscaled to 9 km resolution. No data assimilation is used in ERA5-Land. Therefore, the snowpack evolution in ERA5-Land is solely determined by the atmospheric variables (temperature, snowfall etc.) obtained as forcing from ERA5, and the extensive assimilation of in-situ and remote sensing observations in ERA5 only affects it by constraining these atmospheric input variables. Importantly, the lack of data assimilation in ERA5-Land ensures that there are no artificial sources or sinks of snow. Monthly means of surface air (2 m) temperature, total precipitation, snowfall, snow depth and SWE in a regular

0.1° latitude-longitude grid are used. ERA5-Land data are currently available for 39 winter seasons, from 1981/82 to
       2019/20.

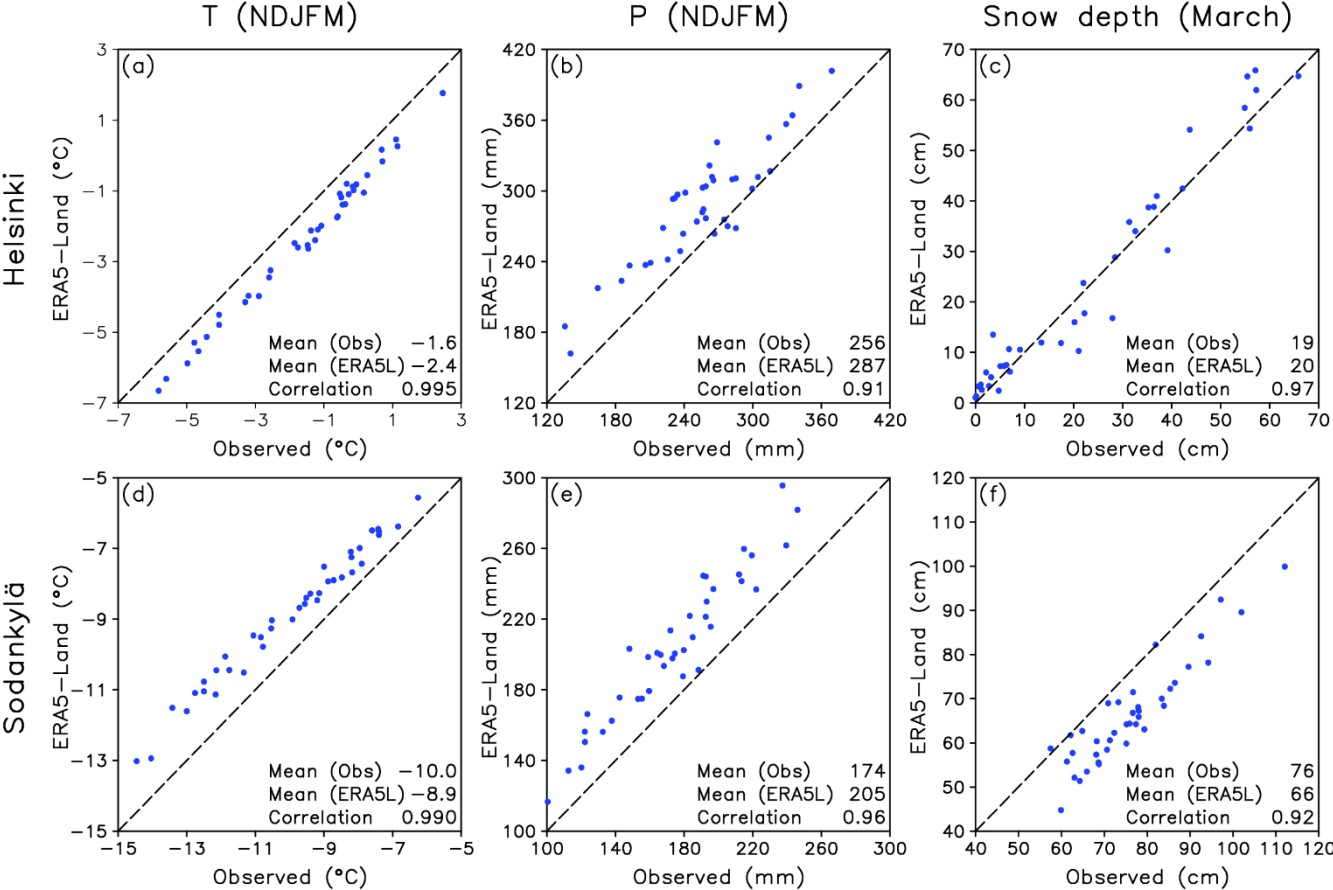

**Figure 3**. Comparison between station observations and ERA5-Land reanalysis of (left) NDJFM mean temperature, (middle) NDJFM
precipitation and (right) March mean snow depth in winters 1981/82-2019/20. Top: station Helsinki Kaisaniemi vs. ERA5-Land at (60.2°
N, 25.0° E); bottom: station Sodankylä Tähtelä vs. ERA5-Land at (67.4° N, 26.6° E). Station observations from the Finnish
       Meteorological Institute (https://en.ilmatieteenlaitos.fi/download-observations).

       ERA5-Land is still a new data set, and the few studies that have already documented some aspects of its performance (e.g.,
       Cao et al., 2020; Pelosi et al., 2020) have not focused on northern Europe. In Fig. 3, ERA5-Land is therefore compared with
station observations at the two locations (Helsinki and Sodankylä) that will be studied in most detail in this paper. The
       interannual variations in the winter season (NDJFM) mean temperature are reproduced with high fidelity (r ≥ 0.99), although
       with a slight negative (positive) bias relative to the local station observations at Helsinki (Sodankylä) (Figs. 3a,d). Given the
       assimilation of surface air temperature observations in ERA5 (Hersbach et al., 2020), this good agreement is perhaps
       unsurprising. The biases may be affected by local factors, such as the urban heat island in the city of Helsinki. Although
ERA5 does not assimilate precipitation measurements in Europe, the interannual correlation is also high (0.91-0.96) for

NDJFM mean precipitation (Figs. 3b,e). The mean values in the reanalysis exceed the station measurements by 12% in Helsinki and 18% in Sodankylä, but this is reasonable considering rain gauge undercatch, which affects especially the measurement of solid precipitation (Adam and Lettenmeier, 2003; Ungersböck et al., 2001). In fact, the difference between ERA5-Land and the station observations agrees well with Taskinen and Söderholm (2016), who estimate the average

December-to-March precipitation in Finland and its cross-boundary watersheds in 1982-2011 to have been 17.5% larger than measured. A similarly high correlation (0.92-0.97) is also found for March mean snow depth (Figs. 3c,f), despite a negative bias in Sodankylä. Such a high agreement for snow depth is remarkable given the mentioned lack of data assimilation in ERA5-Land.

The comparison presented in Fig. 3 is far from exhaustive. More insight could be gained, for example, by extending the evaluation to the daily time scale, but this is out of the focus of the present study. Another unverified aspect is the ability of ERA5-Land to distinguish between solid and liquid precipitation in near-zero temperatures. This is important because, in principle, a good simulation of snow amount might still hide compensating errors in snowfall and snowmelt. Unfortunately, there is no ground truth to compare with, since precipitation measurements in Finland only record the total amounts.

Empirical estimates for the dependence of the snowfall/rainfall probability on near-surface temperature and humidity have been derived based on synoptic observations (e.g., Auer, 1974; Koistinen et al., 2004), but the conversion to total daily snowfall or rainfall fractions is nontrivial because precipitation intensity, temperature and humidity all vary on sub-daily time scales.

## 2.2 EURO-CORDEX-11 RCM simulations

Projected future changes in snow conditions are studied in Sections 6-7 using the EURO-CORDEX-11 RCM simulations (Jacob et al., 2014; Kotlarski et al., 2014). Based on data availability, 17 RCM simulations using boundary conditions from five global climate models (GCMs) were selected for analysis (Table 1). For each GCM-RCM combination, continuous monthly time series of temperature, precipitation, snowfall and SWE were obtained by concatenating the historical simulations (up to the year 2005) with simulations for the RCP8.5 scenario (van Vuuren et al., 2011) for the rest of the 21st

century. RCP8.5 was chosen as the scenario for which the largest number of simulations are available in the EURO-CORDEX-11 data base. However, as a high-end forcing scenario reaching a carbon dioxide concentration of 935 ppm by the year 2100, RCP8.5 may exaggerate the magnitude and rate of climate changes. The EURO-CORDEX-11 simulations were run in a rotated $0.11°$ (ca. 12.5 km) latitude-longitude grid, and their output was re-gridded to a regular $0.1°$ latitude-longitude grid using the nearest neighbour approach. Three 39-winter periods were chosen for analysis: 1981/82 to 2019/20,

2020/21 to 2058/59 and 2059/60 to 2097/98.

**Table 1.** The RCM simulations used in this study.

| Driving GCM | RCM | Institution |
|---|---|---|
| EC-Earth | HIRHAM5 | DMI |
| | RACMO22E | KNMI |
| | RCA4 | SMHI |
| IPSL-CM5A-MR | WRF381P | IPSL |
| | RACMO22E | KNMI |
| HadGEM2-ES | ALADIN63 | ETHZ |
| | WRF381P | IPSL |
| | HadREM3-GA7-05 | MOHC |
| MPI-ESM-LR | COSMO-crCLIM-v1-1 | CLMcom-ETH |
| | HIRHAM5 | DMI |
| | RACMO22E | KNMI |
| | REMO2009 | MPI-CSC |
| NorESM1-M | COSMO-crCLIM-v1-1 | CLMcom-ETH |
| | REMO2015 | GERICS |
| | WRF381P | IPSL |
| | RACMO22E | KNMI |
| | RCA4 | SMHI |

The first column indicates the driving global climate model, the second the regional climate model and the third the

institution that conducted the simulations, using model and institution acronyms at https://esgf-data.dkrz.de/search/cordex-dkrz/.

## 3. Methods

Following Räisänen (2008), the monthly snowfall is written as *FP*, where *P* is the monthly precipitation and *F* is the fraction of precipitation that falls as snow. SWE then becomes


$$SWE = G \int FP dt \qquad\qquad (1)$$

where *G* is the fraction of accumulated snowfall that survives on ground without melting (the snow-on-ground fraction). The time integral of snowfall (*FP*) is evaluated from August to the month considered, but with half-weight for the last month

because the SWE data used in the analysis represent monthly means rather than end-of-month values. *G* is then obtained by dividing SWE by the accumulated snowfall. All the variables required in Eq. (1) (i.e., the total precipitation, snowfall and SWE) are directly available for both ERA5-Land and the EURO-CORDEX simulations.

Denoting the values of $X = SWE$, $G$, $F$ and $P$ in two data samples with subscripts 1 and 2, the mean of $X_1$ and $X_2$ as $\bar{X}$, and the difference $X_2 - X_1$ as $\Delta X$, one obtains

$$\Delta SWE = \underbrace{\bar{G} \int \bar{F} \Delta P dt}_{\Delta SWE(\Delta P)} + \underbrace{\bar{G} \int \Delta F \bar{P} dt}_{\Delta SWE(\Delta F)} + \underbrace{\Delta G \int \bar{F} \bar{P} dt}_{\Delta SWE(\Delta G)} + \underbrace{\frac{1}{4} \Delta G \int \Delta F \Delta P dt}_{\Delta SWE(NL)} \quad (2)$$

Thus, the difference in SWE is decomposed to contributions from the differences in total precipitation ($\Delta P$), snowfall fraction ($\Delta F$) and the snow-on-ground fraction ($\Delta G$), plus a non-linear term that is typically much smaller than the first three right-hand-side terms in Eq. (2). As in Eq. (1), the time integrals in Eq. (2) start from August. The four right-hand-side (rhs) terms in Eq. (2) therefore integrate the effect of weather conditions from August until the month of interest (e.g., March), although the first month that matters in practice is the first month with non-zero mean snowfall. Thus, although the NDJFM season is used for characterizing the winter temperature and precipitation in some of the figures, the diagnostic analysis also uses data outside of this season.

In this study, the decomposition (2) is applied in two different ways:

1. When studying interannual variations in SWE, $X_1$ as defined above Eq. (2) represent the mean values for a 39-winter period (1981/82 to 2019/20, 2020/21 to 2058/59 or 2059/60 to 2097/98) and $X_2$ the values for an individual winter.

2. When studying long-term changes in SWE, $X_1$ represent the mean values for winters 1981/82 to 2019/20, and $X_2$ those for either 2020/21 to 2058/59 or 2059/60 to 2097/98.

Multiplying Eq. (2) with $\Delta SWE$ and averaging over a 39-winter period, the interannual variance of SWE can be decomposed to the contributions of the four right-hand-side (rhs) terms in Eq. (2) as

$$var(SWE) = \langle \Delta SWE^2 \rangle = \langle \Delta SWE \sum_{i=1}^4 \Delta SWE_i \rangle = \sum_{i=1}^4 cov(\Delta SWE_i, SWE) \quad (3)$$

where the angle brackets indicate a time mean, $var$ is variance and $cov$ covariance. Similarly, the standard deviation ($s$) of SWE is decomposed as

$$s(SWE) = \frac{var(SWE)}{s(SWE)} = \sum_{i=1}^4 \frac{cov(\Delta SWE_i, SWE)}{s(SWE)} = \sum_{i=1}^4 \frac{cov(\Delta SWE_i, SWE)}{s(SWE)s(\Delta SWE_i)} s(\Delta SWE_i) \quad (4)$$

which can be rewritten using the definition of correlation ($r$) as

$$s(SWE) = \sum_{i=1}^{4} sdc_i = \sum_{i=1}^{4} r(\Delta SWE_i, SWE)s(\Delta SWE_i) \qquad\qquad (5)$$

where the $sdc_i$:s refer to the *standard deviation contributions* of the four rhs terms in Eq. (2).

Prior to this calculation, all the data are detrended to separate interannual variability from long-term climate change during the 39-year period.

## 4. Illustration of the SWE anomaly decomposition: winters 2010/11 and 2019/20

The use of the decomposition (2) is illustrated for two winters (2010/11 and 2019/20) in Fig. 4, using the ERA5-Land grid boxes closest to Helsinki (60.2° N, 25.0° E) and Sodankylä (67.4° N, 26.6° E) (H and S in Fig. 1). These winters had very different snow conditions. In Helsinki, winter 2019/20 was nearly snow-free, but the SWE in 2010/11 was well above the average (Fig. 4e; see also Lehtonen, 2015). By contrast, the SWE in Sodankylä was record large in 2019/20 but below the average in 2010/11 (Fig. 4l). Winter 2010/11 was cold in both Helsinki and Sodankylä (Figs. 4a,h), whereas winter 2019/20 was record mild in Helsinki. It was also milder than the average in Sodankylä where, however, both October-November and April-May were relatively cold.

Figs. 4b-d (Helsinki) and 4i-k (Sodankylä) show the three factors that regulate SWE based on Eq. (1). At both locations, total precipitation was larger in 2019/20 than in 2011/20 (Figs. 4b,i). In Helsinki, however, the snowfall fraction $F$ (Fig. 4c) and particularly the snow-on-ground fraction $G$ (Fig. 4d) were very low in 2019/20 but much higher in 2010/11, reflecting the very different temperatures in these two winters. In the colder climate of Sodankylä, $F$ and $G$ differed much less between these two winters (Figs. 4j,k). Still, $G$ was systematically higher in Sodankylä in 2019/20. This is explained by two factors. First, although the period from November to March was much milder in 2019/20 than 2010/11, October and April-May were colder. The colder temperatures in these months reduced early and late season snowmelt in 2019/20, whereas the warm anomalies in the middle of the winter had little effect because the temperature in Sodankylä still mostly remained well below zero. Second, because of the more abundant precipitation in 2019/20 than 2010/11, there was also more snowfall. Thus, to have the same value of $G$ in these two winters, a larger absolute amount of snowmelt would have been needed in 2019/20.

In Helsinki, decomposition (2) attributes both the positive SWE anomalies in 2010/11 and the negative SWE anomalies in 2019/20 mostly to anomalies in the snow-on-ground fraction $G$, with a secondary contribution from anomalies in the snowfall fraction $F$ (Figs. 4f-g). In Sodankylä, anomalies in total precipitation dominate in both winters until April (Figs. 4m-n). However, most of the negative (positive) SWE anomaly in May 2011 (2020) is attributed to a lower (higher) than average snow-on-ground fraction.

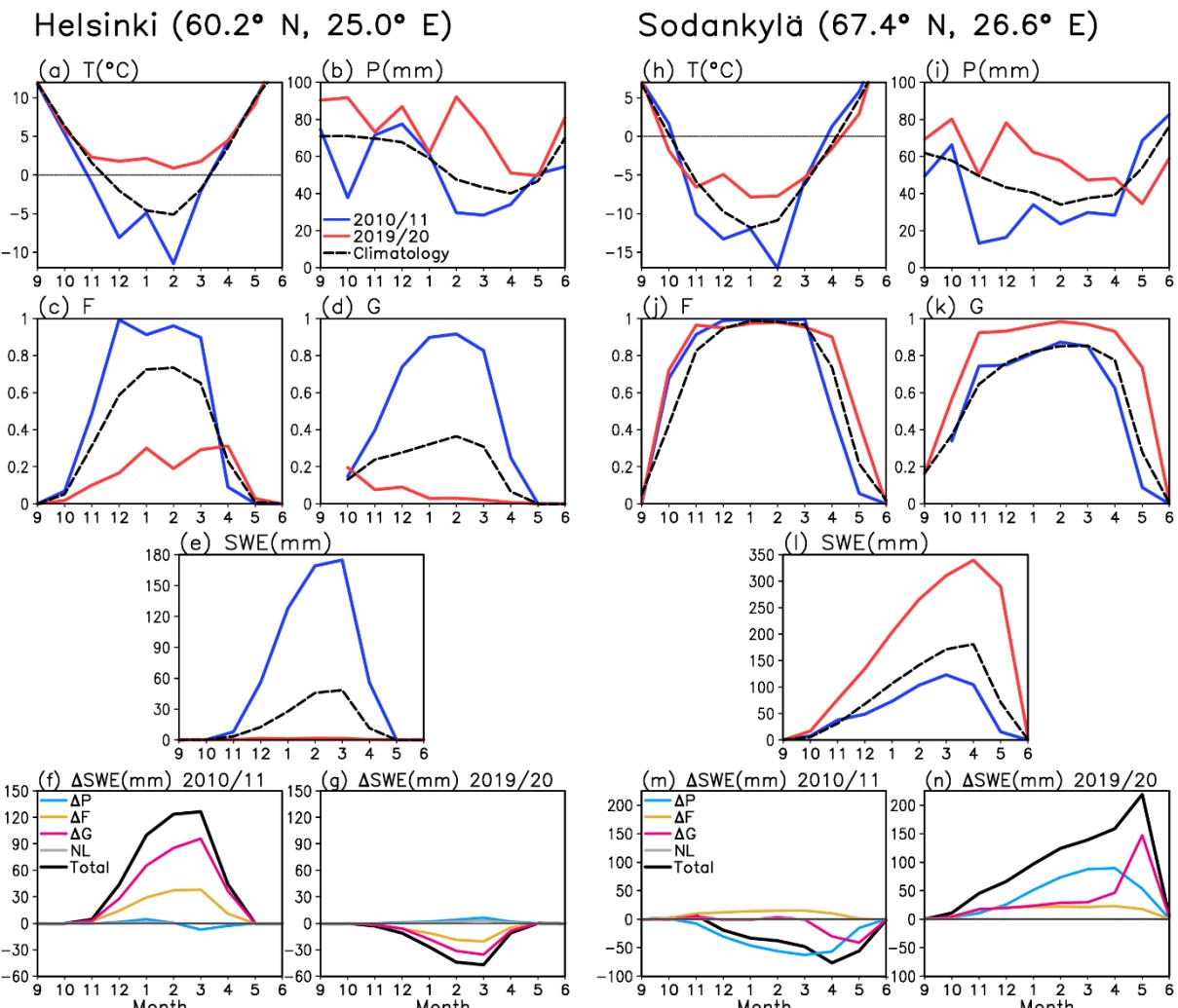

**Figure 4.** (a)-(g) Weather and SWE diagnostics for winters 2010/11 and 2019/20 for the ERA5-land grid box closest to Helsinki. (a)-(e): temperature T, precipitation P, snowfall fraction F, snow-on-ground fraction G and SWE in 2010/11 (blue), 2019/20 (red) and the corresponding 39-winter mean values (black). (f)-(g): decomposition of the winter 2010/11 and 2019/20 SWE anomalies to contributions from the four rhs terms in (2) (see the legend in (f) for line colours). (h)-(n) The same for Sodankylä.

## 5. Interannual SWE variability in ERA5-Land

Time series of SWE anomalies in March 1982-2020 are shown in Fig. 5 for the ERA5-Land grid points closest to Helsinki and Sodankylä. Although the average SWE is much smaller in Helsinki than in Sodankylä, the interannual variability of SWE (shown by the solid line) is larger in Helsinki, signifying much more irregular snow conditions. The decomposition (2) identifies the snow-on-ground fraction (red bars in Fig. 5a) as the dominant source of variability in Helsinki. Variations in the snowfall fraction (yellow bars) also tend to amplify the SWE anomalies, whereas anomalies in total precipitation (blue

bars) either reinforce (e.g. 1984) or oppose (e.g. 1996) the actual SWE anomaly. The non-linear term (grey bars) is generally

negligible.

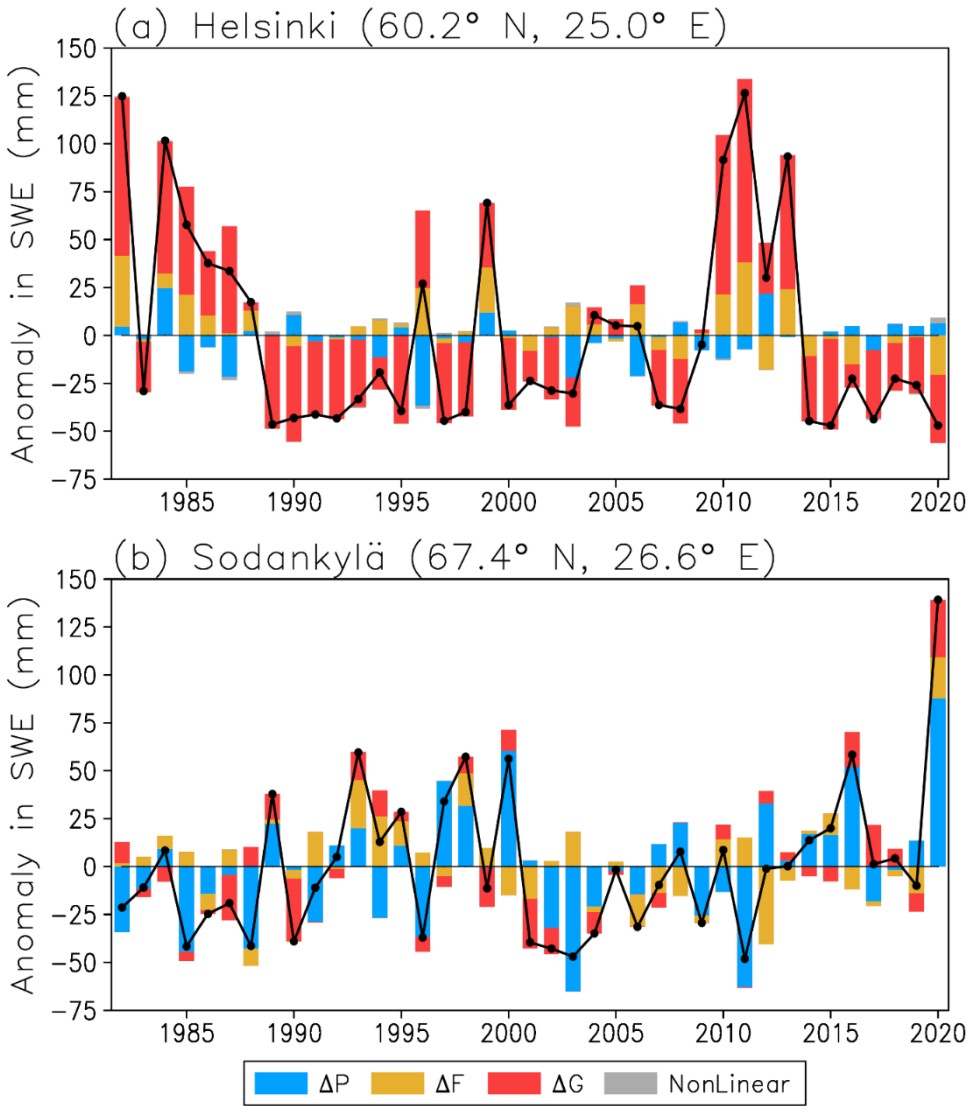

**Figure 5.** Anomalies of March mean SWE in (a) Helsinki and (b) Sodankylä in years 1982-2020 based on ERA5-Land. The solid line shows the total SWE anomaly and the bars the contributions of the four rhs terms in Eq. (2).

In stark contrast with Helsinki, the interannual variations of March mean SWE in Sodankylä (Fig. 5b) are in most years

dominated by anomalies in total precipitation. Variations in snowfall fraction and the snow-on-ground fraction play a smaller and less systematic role. Decomposition of the standard deviation of March mean SWE using Eq. (5) confirms the visual impression from these time series (Table 2).

**Table 2.** Standard deviation (mm) of detrended March mean SWE anomalies in years 1982-2020 decomposed to its contributions from the four rhs terms in Eq. (2). The values in parentheses give the standard deviations of the individual terms ($s(\Delta SWE_i)$ in Eq. (5)) and their correlation with the SWE anomaly ($r(\Delta SWE_i, SWE)$ in Eq. (5)).

|  | Helsinki | Sodankylä |
|---|---|---|
| Total precipitation ($\Delta P$) | 0 (11, -0.01) | 26 (31, 0.84) |
| Snowfall fraction ($\Delta F$) | 10 (13, 0.75) | 4 (13, 0.30) |
| Snow-on-ground fraction ($\Delta G$) | 41 (42, 0.97) | 7 (12, 0.59) |
| Non-linear | 0 (1, -0.35) | 0 (0.1, -0.14) |
| SWE | 51 | 37 |

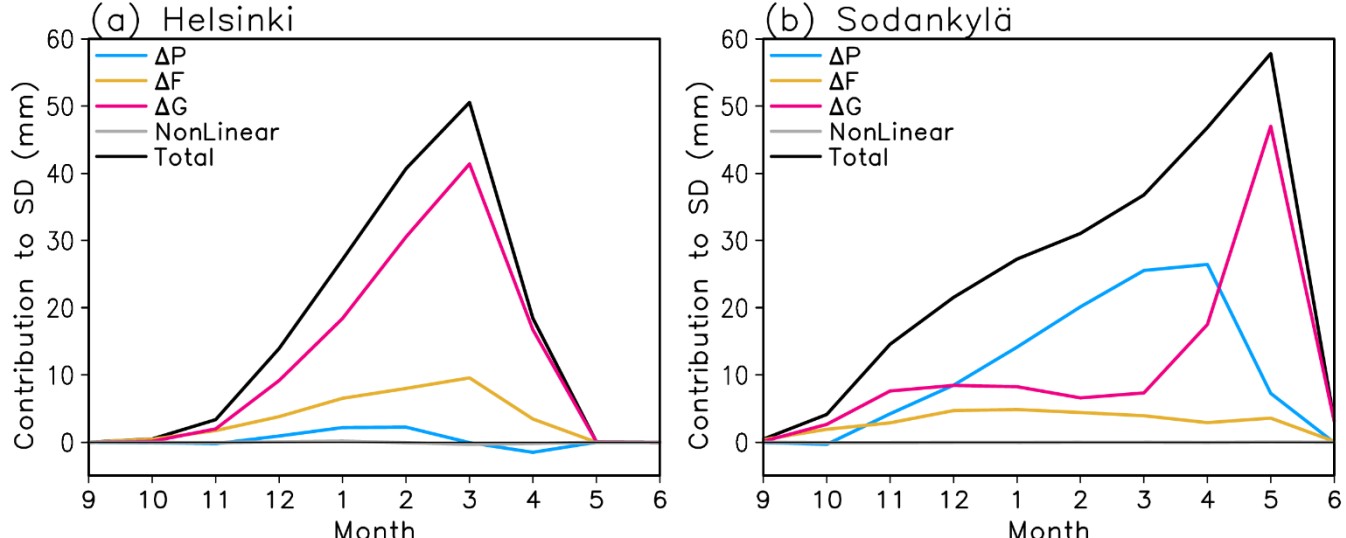

**Figure 6.** (a) Interannual standard deviation of SWE (black line) in Helsinki and the contributions of the four rhs terms in Eq. (2) to it (coloured lines) based on ERA5-Land. (b) The same for Sodankylä.

Figure 6 shows how the resulting standard deviation contributions in Helsinki and Sodankylä evolve during the winter season. In Helsinki (Fig. 6a), the standard deviation of SWE increases quasi-linearly from November to March, when the mean SWE also reaches its maximum (Fig. 4e). Variations in the snow-on-ground fraction (red line) dominate the SWE variability throughout the snow season, with a secondary contribution from the snowfall fraction (yellow line). In Sodankylä (Fig. 6b), variations in total precipitation (blue line) provide the largest contribution to SWE variability from January to April. However, the snow-on-ground fraction becomes increasingly variable during the melting season in spring, dominating the SWE variability in May. It also makes the largest contribution to the SWE variability in Sodankylä in October and November. Thus, SWE in both the beginning and the end of the snow season is less sensitive to the total precipitation and the snowfall fraction than to the fraction of the accumulated snowfall that survives on ground.

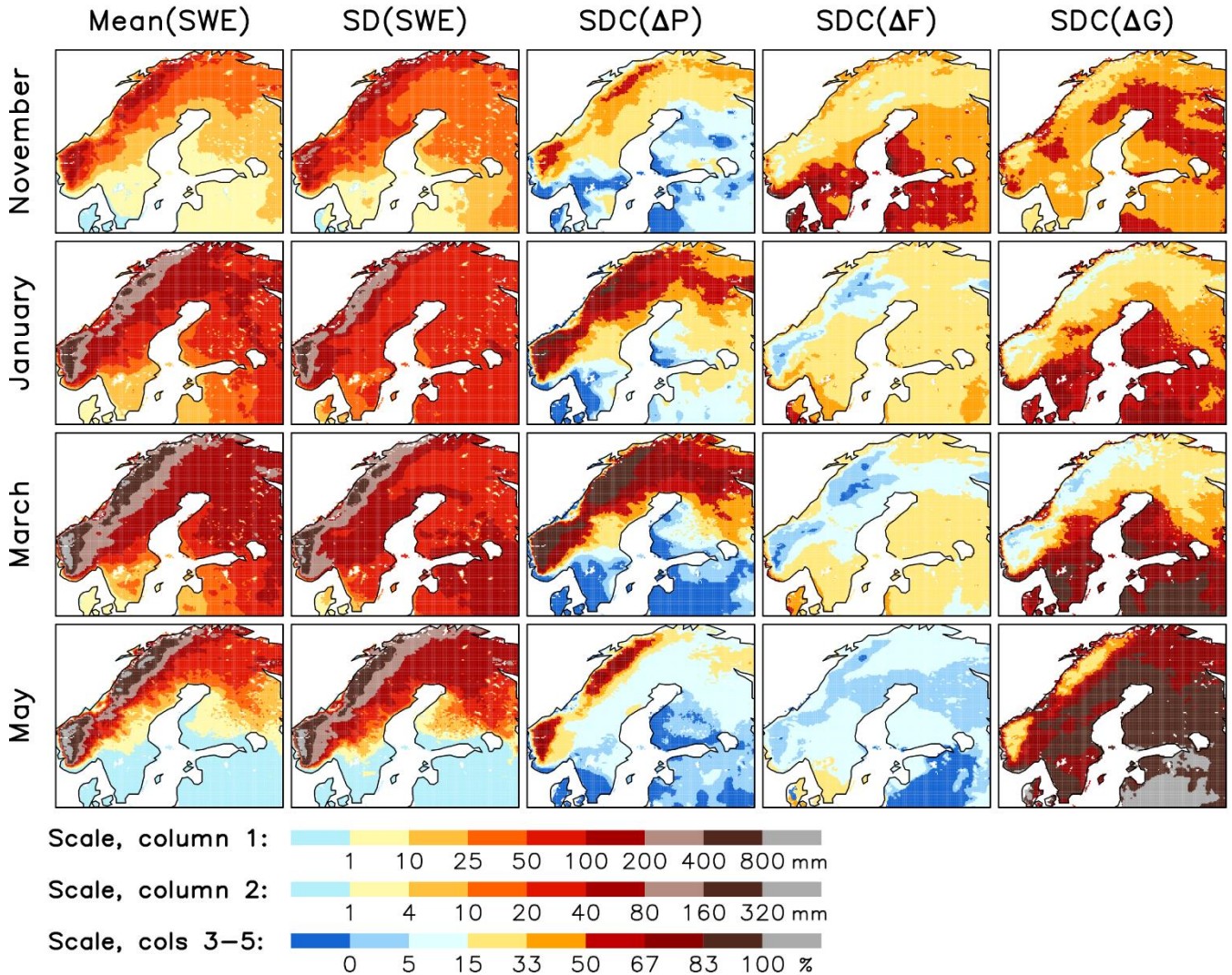

**Figure 7.** Statistics of SWE in ERA5-Land, winters 1981/82 to 2019/20. Columns 1-2: mean and interannual standard deviation of SWE (in mm). Columns 3-5. Relative contributions of total precipitation (ΔP), snowfall fraction (ΔF) and snow-on-ground fraction (ΔG) to the standard deviation of SWE (in %). The colour scales are given at the bottom of the figure.

A wider perspective of the mean SWE and its interannual variability in northern Europe is provided in Fig. 7. The mean SWE for winters 1981/82 to 2019/20 shows a strong maximum over the Scandinavian mountains where precipitation is abundant and winters are long (column 1). In most of the area, SWE is close to its maximum in March, although the exact time varies between January (in Denmark) and May (mountains in northern Sweden). The milder parts of the domain up to southern Finland are practically snow-free in May. The interannual standard deviation of SWE follows broadly the same

geographical pattern (column 2). As one exception, the standard deviation in March is larger in Estonia and southern Finland than in the Finnish Lapland, although the mean SWE is much smaller (see also Figs. 4-5). The coefficient of variation of

SWE (standard deviation divided by mean), which represents the relative irregularity of snow conditions, tends to increase from colder to milder regions (Fig. A1).

The last three columns in Fig. 7 show the relative (per cent) contributions of the three main terms in Eq. (2) to the standard deviation of SWE. Focusing first on the height of the winter in January and March, there is a steep contrast in the drivers of the variability between the colder and milder parts of the domain. In cold areas, including the Scandinavian mountains, northern Sweden and northern Finland, a majority of the SWE variability is associated with variations in total precipitation (column 3). In milder regions such as Denmark, coastal Norway, the Baltic States, and southern-to-central Sweden and

Finland, variations in the snow-on-ground fraction are dominant (column 5). Variations in the snowfall fraction (column 4) also amplify the SWE variability in most areas, although their contribution in January and March is typically smaller than those of the two other terms.

In the beginning of the snow season in November, the SWE variability dynamics is somewhat different (top row of Fig. 7).

The role of total precipitation is smaller than in January and March, whereas variations in the snowfall fraction are more important, explaining more than a half of the SWE standard deviation in some of the milder areas. Compared with January and March, the snow-on-ground fraction explains a smaller percentage of the SWE variability in November in the milder parts of the domain (mirroring the larger share of variations in the snowfall fraction) but a larger percentage in colder areas. In May, variations in the snow-on-ground fraction widely govern the SWE variability in lowland areas of northern Europe,

but the contribution of total precipitation is still dominant over the Scandinavian mountains (bottom row of Fig. 7).

A comparison between Figs. 2 and 7 suggests a strong temperature dependence in the drivers of interannual SWE variability, in the sense that precipitation anomalies become more important, and anomalies in snowfall and snow-on-ground fractions less important, with decreasing mean temperature. Earlier, Mankin and Diffenbaugh (2015) found a similar baseline climate

dependence in the dynamics of interannual SWE variability in a wider geographical context. In their simulations with the CCSM3 model, the March mean SWE was more strongly related to the NDJFM precipitation than temperature in areas such as Siberia and northern Canada, but vice versa in most midlatitude regions including much of northern Europe (their Fig. 6).

As a further illustration, the relative contribution of precipitation variability to SWE variability in March (row 3, column 3 in

Fig. 7) is plotted as a function of the climatological NDJFM mean temperature in Fig. 8a. On the average, this contribution exceeds 80 % where $T_{NDJFM} < -11°C$, is close to 50 % where $T_{NDJFM} \approx -7°C$, and decreases to zero at $T_{NDJFM} \approx -2°C$. Despite the non-linearity of the relationship, there is a strong negative spatial correlation ($r = -0.85$) between the two variables in Fig. 8a. Conversely, the relative contributions of snowfall fraction variability ($sdc(\Delta F)/sdc(SWE)$) and snow-on-ground fraction variability ($sdc(\Delta G)/sd(SWE)$) are positively correlated with the NDJFM mean temperature ($r = 0.65$

and 0.83, respectively).

Nevertheless, the dynamics of interannual SWE variability is not solely controlled by the winter mean temperature. For the same NDJFM mean temperature, $sdc(\Delta P)/sd(SWE)$ tends to increase with increasing NDJFM mean precipitation (see the colour coding in Fig. 8a). In particular, the SWE variability in western Norway, where more precipitation falls than elsewhere in Northern Europe (Fig. 8b), is more strongly affected by precipitation variability than expected from the winter mean temperature alone. On one hand, the larger mean precipitation is associated with larger absolute precipitation variability. On the other hand, larger amounts of snowfall reduce the variability in the snow-on-ground fraction, because a larger amount of snowmelt is needed for a unit change in the latter.

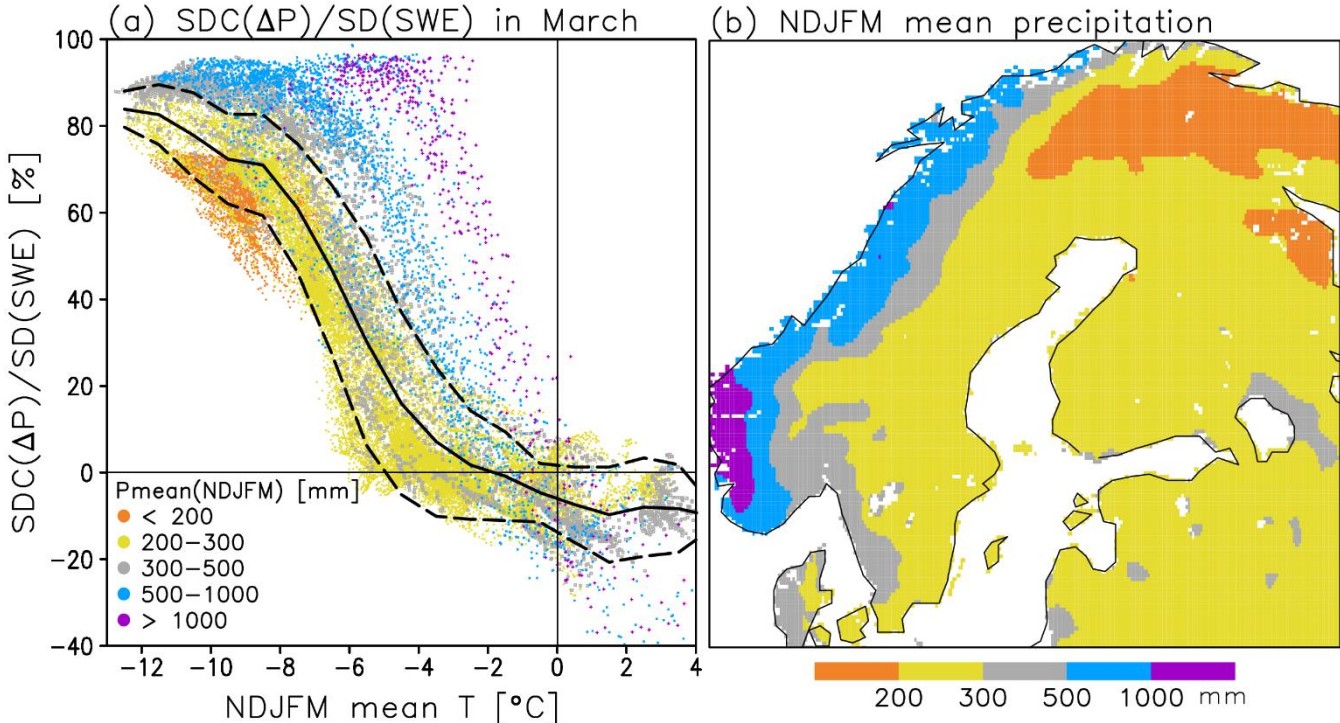

**Figure 8.** (a) The relative contribution of precipitation anomalies to the standard deviation of SWE in March as a function of the NDJFM mean temperature in 1981/82 – 2019/20. Each dot represents a single 0.1° × 0.1° grid cell, coloured according to the mean NDJFM precipitation shown in (b). The solid line in (a) indicates the mean values for 1°C temperature bins, and the two dashed lines the mean ± one standard deviation.

## 6. Future changes in mean SWE in the EURO-CORDEX simulations

We now turn on to the EURO-CORDEX simulations to address two questions related to the projected climate change during the rest of this century. In this section, the focus is on changes in long-term mean SWE. In Section 7, we study how the interannual variability of SWE changes and the processes contributing to its change.

## 6.1 Projected SWE changes and their diagnostic decomposition

A warming climate leads to a simulated decrease in SWE in Finland throughout the winter season (Fig. 9, left column).
Consistent with Räisänen and Eklund (2012), the relative decrease is much larger in Helsinki in the south than in Sodankylä
in the north. In Helsinki, the multi-RCM mean SWE in 1981/82-2019/20 reaches 50 mm in March, in reasonable agreement
with ERA5-Land (Fig. 4e). Later in the 21st century, the maximum shifts earlier to February and decreases by 43% to 29
mm in 2020/21-2058/59 and by 77% to 12 mm in 2059/60-2097/98. The maximum in Sodankylä reaches 160 mm in March
in 1981/82-2019/20, which is slightly below and earlier than the estimate from ERA5-land (Fig. 4l). This then decreases by
17% to 135 mm in 2020/21-2058/59 and by 38% to 100 mm in 2059/60-2097/98. Recall that these simulations are based on
the RCP8.5 scenario. Under a lower trajectory of greenhouse gas emissions, the decrease in SWE would remain smaller.

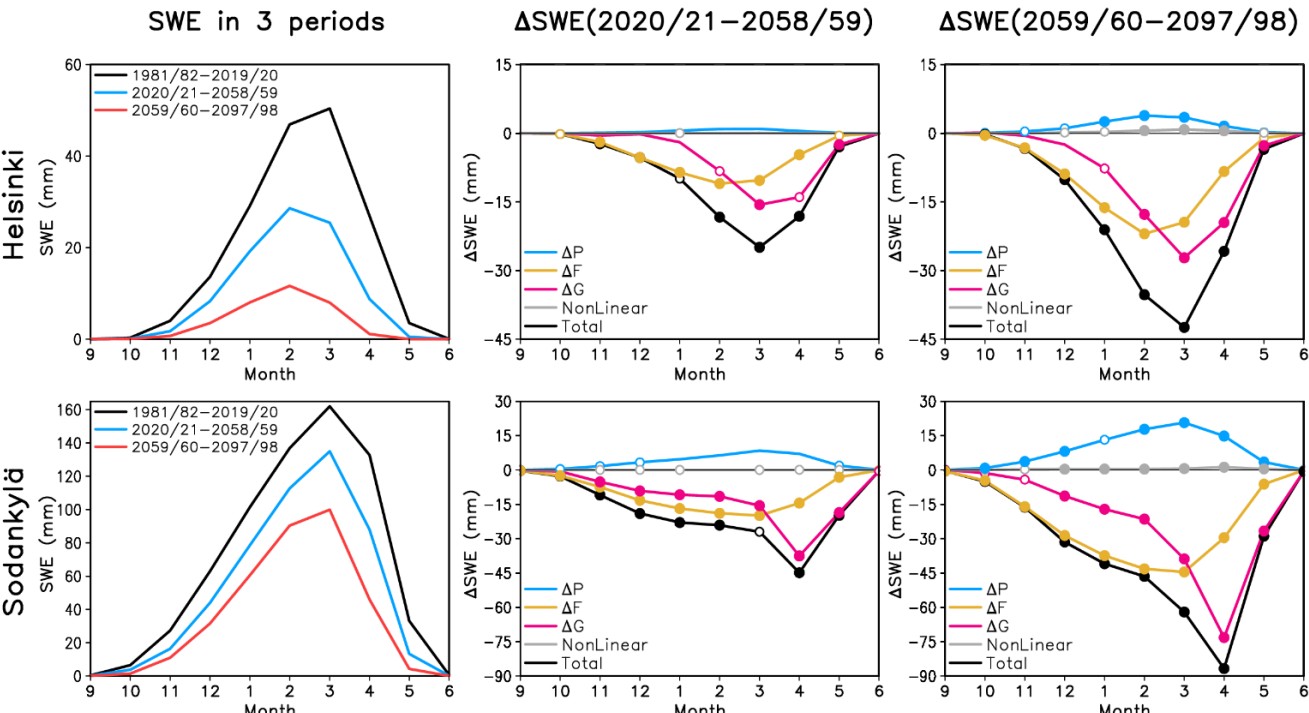

**Figure 9.** Left: multi-RCM mean SWE in Helsinki and Sodankylä in the winters 1981/82-2019/20 (black), 2020/21-2058/59 (blue) and
2059/60-2097/98 (red). Middle: the changes from 1981/82-2019/20 to 2020/21-2058/59 decomposed to the contributions of the four rhs
terms in Eq. (2) (see the legend for line colours). Months in which all 17 (14-16 of the 17) simulations agree on the sign of the change are
indicated with a closed (open) circle. Right: as middle, but for the changes from 1981/82-2019/20 to 2059/60-2097/98.

Diagnosing the causes of the SWE change with Eqs. (1)-(2) reveals dynamics very similar to those documented by Räisänen
and Eklund (2012) for the ENSEMBLES RCMs (van der Linden et al., 2009) (Fig. 9, middle and right columns). There is an
ensemble mean increase in winter precipitation in the EURO-CORDEX simulations which, if acting alone, would increase
the SWE in both Helsinki and Sodankylä. This increase in precipitation is not yet very robust across the 17 individual RCM
simulations in 2020/21-2058/59, but by 2059/60-2097/98 all or nearly all these simulations agree on it (the closed and open

circles in Fig. 9). However, the effect of increasing total precipitation is more than compensated by decreases in the snowfall fraction (yellow lines in the middle and right panels of Fig. 9) and the snow-on-ground fraction (red lines). Until February in Helsinki and until March in Sodankylä, the decrease in the snowfall fraction makes a larger contribution to the SWE change than the decrease in the snow-on-ground fraction. This contrasts with the dynamics of interannual SWE variability, in which variations in the snowfall fraction are mostly secondary to those in the snow-on-ground fraction (Figs. 6-7). However, the decrease in SWE in spring (beginning from March in Helsinki and April in Sodankylä) is dominated by earlier snowmelt and thus decreasing snow-on-ground fraction.

In Figure 10, maps of the multi-RCM mean SWE change are shown for March, which is close to the peak of the snow season in most of northern Europe. The first two columns reveal a decrease in SWE in practically the whole area. As an exception, the sign of the change is locally ambiguous (less than 80%, or 14 out of the 17 simulations agreeing on it) over the coldest parts of the Scandinavian mountains in northern Sweden and southern Norway, particularly in the first future period (2021-2059). This is similar to Räisänen and Eklund (2012), who found the increase in total precipitation to locally overcompensate the decrease in snowfall and snow-on-ground fractions in northern Sweden in the ENSEMBLES RCMs. More generally, the relative decrease in SWE increases from colder to milder regions, i.e. from high to low elevations and from north to south (column 2 of Fig. 10), although the absolute decrease (column 1) is fairly similar across large parts of Sweden and Finland. Obviously, the decrease in SWE is much larger in the second (2060-2098) than in the first (2021-2059) future period.

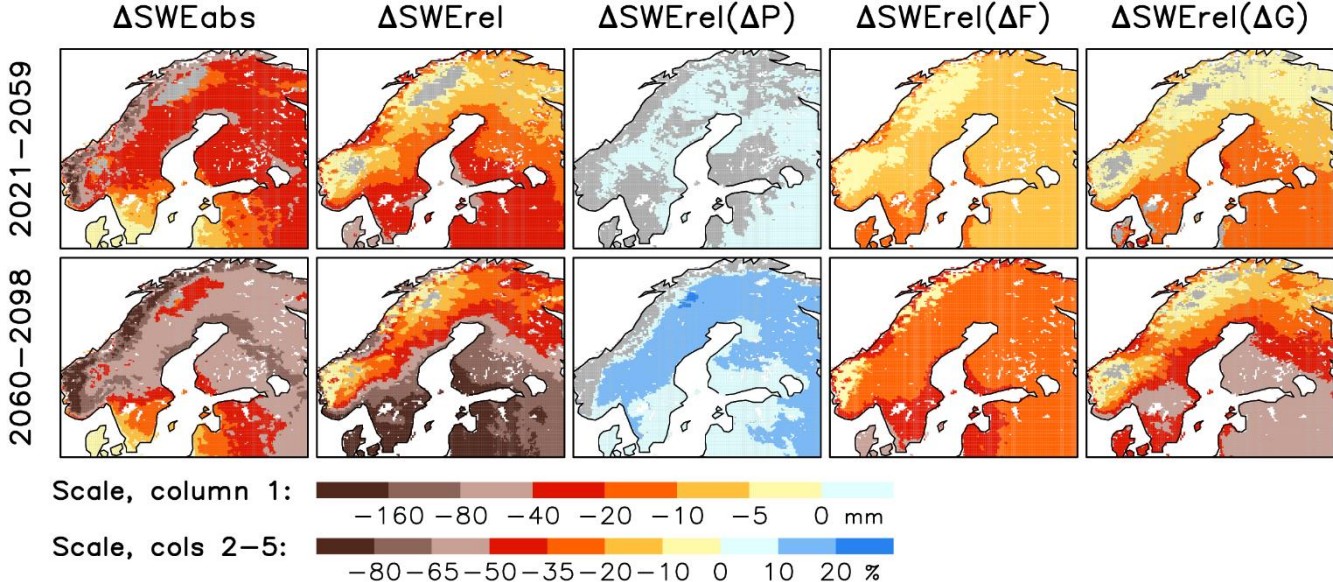

**Figure 10.** Multi-RCM mean changes in March mean SWE from years 1982-2020 to 2021-2059 (top) and 2060-2098 (bottom). Columns 1-2: change in SWE in absolute (mm) units and in per cent of the 1982-2020 multi-RCM mean. Columns 3-5: contributions of total precipitation change (ΔP), snowfall fraction (ΔF) change and snow-on-ground fraction (ΔG) change to the per cent change in SWE. Grey shading is used in areas where less than 14 of the 17 RCM simulations agree on the sign of the change.

As shown by the last three columns in Fig. 10, the dynamics of the SWE change in Helsinki and Sodankylä (Fig. 9) are broadly generalizable to the rest of northern Europe. Increasing total precipitation, if acting alone, would lead to a slight general increase in SWE (column 3 of Fig. 10), although the signal is not very robust across the EURO-CORDEX ensemble in years 2021-2059 and it remains non-robust in western and northern Norway even in 2060-2098. However, decreasing snowfall and snow-on-ground fractions act to reduce the SWE (columns 4-5), and they both contribute to the larger relative

decrease in SWE in mild than cold areas. This geographical contrast is larger for the change in the snow-on-ground fraction, although this partly depends on the month chosen for analysis.

### 6.2 Further discussion of SWE changes: future projections versus interannual variability and observed trends

In apparent conflict with the simulated future decrease in SWE nearly everywhere in northern Europe, Fig. 2a showed a positive interannual correlation between March mean SWE and NDJFM mean temperature over the Scandinavian mountains

and in the northern parts of Sweden and Finland. This conflict arises because the relationship between winter temperature and precipitation differs between the long-term climate change and the interannual variability. As discussed below based on Fig. 11, the projected long-term increase in winter precipitation is in most of northern Europe smaller than the projected warming together with the interannual regression relationship between temperature and precipitation anomalies in ERA5-Land indicates.


The EURO-CORDEX RCMs simulate, on the average, a NDJFM mean warming of ca. 3-5 °C from 1981/82-2019/20 to 2059/60-2097/98, with a general increase from southwest to northeast (Fig. 11a). The change in precipitation varies from slight local decreases in western and northern Norway to increases of up to 25 %, with a relatively sharp northwest-to-southeast contrast across the Scandinavian mountains (Fig. 11b). This contrast is qualitatively similar to that found by

Räisänen and Eklund (2012), but its connection to the atmospheric circulation in the EURO-CORDEX RCMs would require further investigation. The multi-RCM mean changes in the NDJFM mean sea level pressure in northern Europe are small (from 0 to +1 hPa), implying only very modest changes in the average lower tropospheric winds (not shown).

The ratio between the precipitation and temperature changes is mostly 2-6 % $(°C)^{-1}$, but lower in western and northern

Norway (Fig. 11c). On the interannual time scale, however, a 1 °C positive temperature anomaly is statistically accompanied by a 12-15 % precipitation anomaly in western Norway (Fig. 11d), where westerly flow anomalies result both in advection of warm Atlantic air and forced ascent uphill the Scandinavian mountains. The interannual regression coefficient (Fig. 11d) also exceeds the long-term precipitation-to-temperature change ratio (Fig. 11c) in Finland and northern Sweden. For example, in the grid box closest to Sodankylä, the long-term change ratio (3.4 % $(°C)^{-1}$) is only half of the interannual slope

(6.1 % $(°C)^{-1}$) in ERA5-Land. The interannual regression coefficients in the EURO-CORDEX RCMs agree generally well with ERA5-Land (not shown).

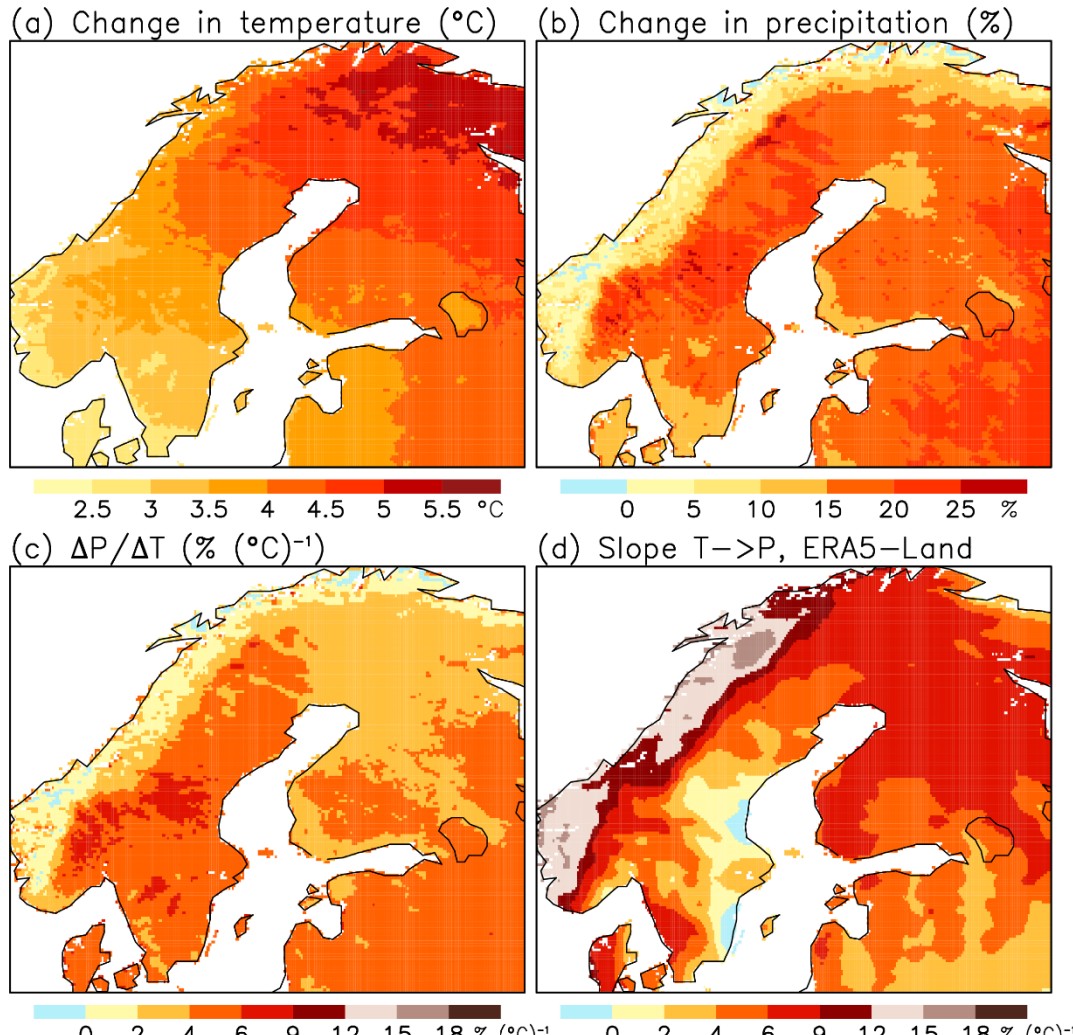

**Figure 11.** Multi-RCM mean changes in NDJFM mean (a) temperature $T$ and (b) precipitation $P$ from 1981/82-2019/20 to 2059/60-2097-98, and (c) the ratio of the precipitation change to the temperature change. (d) slope $b$ in least-square linear regression $P(\text{NDJFM}) = a + bT(\text{NDJFM})$ for interannual variability in 1981/82-2019/20, in ERA5-Land.

Thus, while long-term climate change accords qualitatively with interannual variability in the sense that winter precipitation increases together with temperature, there is no quantitative analogy. The long-term precipitation increase is in most of northern Europe smaller, and the ability of increased precipitation to compete with reduced snowfall and snow-on-ground fractions is therefore weaker, than the interannual relationship would suggest. This explains why SWE decreases in nearly the whole northern Europe, despite the positive interannual temperature-SWE correlation in a significant part of the area.

A caveat in any model-based analysis is that climate changes in the real world may or may not follow the model projections. Interestingly, despite a decrease in winter mean and maximum snow depth in large parts of Europe since the 1950s

(Fontrodona Bach et al., 2018), Skaugen et al. (2012) found generally positive trends in winter maximum SWE above the 850 m altitude in southern Norway in the period 1931-2009. On a larger scale, Zhong et al. (2018) analysed observations of winter maximum snow depth in the Former Soviet Union, Mongolia and China, finding an average positive trend of 0.6 cm decade$^{-1}$ from 1966 through 2012. Increases in snow depth dominated especially north of 50°N, extending to milder regions than one would expect based on GCM projections for the future (Räisänen, 2008). Whether such differences reflect a problem in the models or have resulted from multidecadal internal variability in the atmospheric circulation (Deser et al., 2012; Mankin and Diffenbaugh, 2015) is still an open question. If the atmospheric circulation turned out to be more sensitive to increasing greenhouse gas concentrations than current climate models indicate (as tentatively suggested by Scaife and Smith, 2018), some of the present conclusions might need to be modified.

## 7. Future changes in SWE variability in the EURO-CORDEX simulations

Identically to the processing of the ERA5-Land data, the interannual standard deviation of SWE in the EURO-CORDEX simulations was calculated from detrended 39-winter time series separately for the periods 1981/82-2019/20, 2020/21-2058/59 and 2059/60-2097/98, and the contributors to the SWE variability were diagnosed using Eqs. (2) and (5). Figure 12 shows the results for the grid boxes closest to Helsinki and Sodankylä. In the near-present period 1981/82-2019/20 (left column), the model results agree reasonably well with ERA5-Land (Fig. 6). In particular, the SWE variability in Helsinki is largely driven by variations in the snow-on-ground fraction. In Sodankylä, variations in total precipitation make the largest contribution from January to March, although this term is not as clearly dominant as in ERA5-Land (Fig. 6b). The magnitude of the standard deviation is also comparable to ERA5-Land, although slightly smaller in Helsinki nearly throughout the winter and in Sodankylä in May.

Later during the 21st century, the interannual standard deviation of SWE decreases in Helsinki (top middle and right in Fig. 12), reflecting the large decrease in the average SWE. However, the decrease in the standard deviation is in per cent terms smaller than the decrease in the mean; for example, by 2059/60-2097/98 the winter maximum of monthly mean SWE decreases by 77 % whereas the maximum of the standard deviation decreases by 65 %. This suggests that the snow conditions are becoming increasingly irregular, with an increasing number of virtually snow-free winters but a smaller relative decrease in SWE in the most snow-rich winters than in an average winter. Apart from an increasing frequency of midwinter snowmelt events, this likely reflects an increase in relative snowfall variability as the number of days with snowfall decreases but the intensity of the largest snowfall events remains nearly unchanged (O' Gorman, 2014; Räisänen, 2016).

The standard deviation of SWE also decreases in Sodankylä, but much less than in Helsinki. Following the earlier snowmelt in a warmer climate, the maximum of the standard deviation shifts from April to March in the last 39-year period (2060-

2098). Note, though, that the standard deviation of SWE in Sodankylä in years 1982-2020 reaches its maximum earlier in the RCMs than in ERA5-Land (bottom left of Fig. 12 vs. Fig. 6b), just as the mean SWE does (bottom left of Fig. 9 vs. Fig. 4l). This bias naturally affects the quantitative interpretation of the model projections.

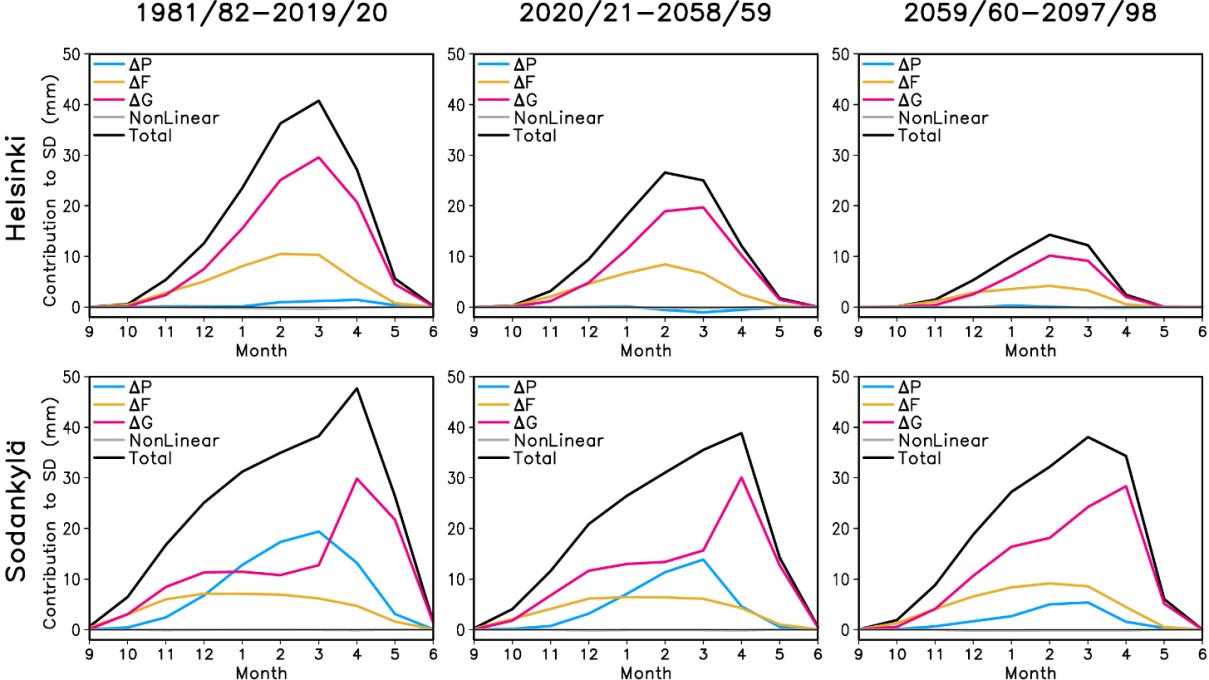

**Figure 12.** Multi-RCM mean interannual standard deviation of SWE (black line) in Helsinki (top) and Sodankylä (bottom) in the periods 1981/82-2019/20 (left), 2020/21-2058/59 (middle) and 2059/60-2097/98 (right) and the contributions of the four rhs terms in Eq. (2) to it (coloured lines).

In Helsinki, variations in the snow-on-ground fraction are the dominant driver of interannual SWE variability in all three periods, with a secondary contribution from the variation in the snowfall fraction. In Sodankylä, however, a systematic change in the drivers of SWE variability is seen. Variations in total precipitation become gradually less important with time, whereas variations in the snow-on-ground fraction and (secondarily) the snowfall fraction become more important. In the last 39-year period, variation in the snow-on-ground fraction is the largest driver of SWE variability in Sodankylä from December to the end of the snow season.

The maps in the top row of Fig. 13 show the relative contributions of total precipitation, snowfall fraction and snow-on-ground fraction to the standard deviation of March mean SWE in years 1982-2020, as averaged over the 17 RCM simulations. Comparison with ERA5-Land (row 3 in Fig. 7) reveals a good agreement on the main geographical patterns. The SWE variability over the Scandinavian mountains and in much of Swedish and Finnish Lapland is mainly driven by precipitation variability, whereas variations in the snow-on-ground fraction dominate the variability in lowland areas further south. Quantitatively, the gradient between the precipitation and snow-on-ground fraction dominated zones is less steep for

the multi-RCM mean than for ERA5-Land (compare, for example, the difference between southwestern and Northern Finland in row 1 of Fig. 13 and row 3 of Fig. 7). This smoothing of gradients results at least partly from averaging over multiple RCM simulations with somewhat different climates.

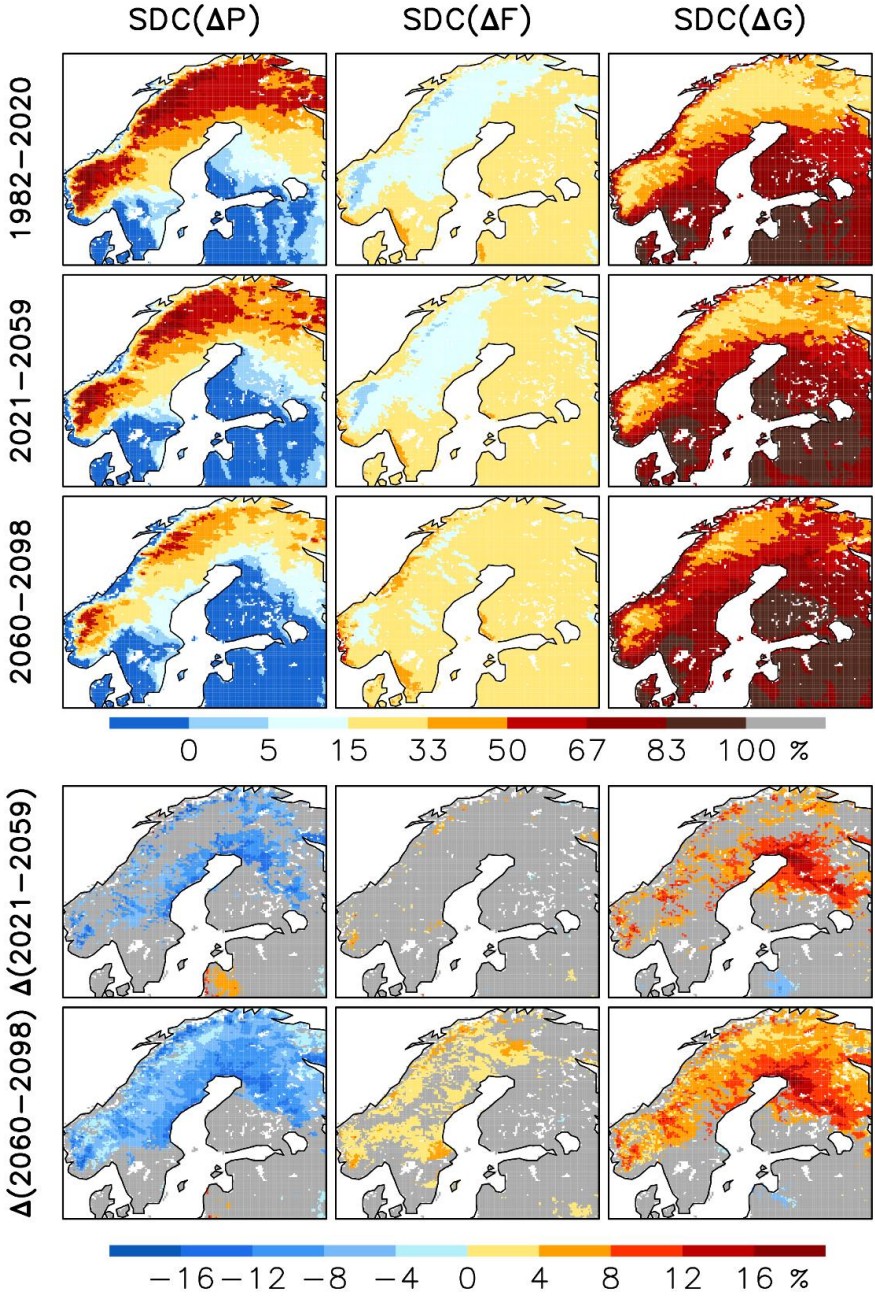

**Figure 13.** Relative contributions of variations of total precipitation (ΔP), snowfall fraction (ΔF) and snow-on-ground fraction (ΔG) to the multi-RCM mean of the standard deviation of March mean SWE in the EURO-CORDEX simulations. Rows 1-3: the contributions in three 39-year periods separately. Rows 4-5: the changes from 1982-2000 to 2021-2059 and 2060-98. Grey shading is used in areas in which less than 14 of the 17 RCMs agree on the sign of the change.

Reflecting the warming of winters later in the 21st century, the variability in total precipitation tends to become a smaller contributor to the SWE variability, whereas the variations in the snow-on-ground fraction and to a lesser extent the snowfall fraction become relatively more important (rows 2-5 of Fig. 13). The change relative to 1982-2020 is still fairly subtle in most parts of Northern Europe in 2021-2059, as indicated by the relatively small fraction of areas in which more than 80% of the EURO-CORDEX RCMs agree on its sign (row 4). However, the signal grows stronger by 2060-2098, when there is

widespread agreement on the reduced importance of precipitation variability in those areas where it is important in the near-present climate (rows 3 and 5). Similarly, there is good agreement on the increased role of snow-on-ground fraction variability in broadly the same areas. Variations in the snowfall fraction also tend to become more important, although good intermodel agreement on this is mostly confined to scattered areas in central-to-northern Norway and Sweden.

The changes seen in Fig. 13 follow the expectations raised by the present-day geographical contrasts in the mechanisms of interannual SWE variability (Fig. 7). As SWE variability tends to be mostly driven by variations in winter total precipitation in sufficiently cold climates and by variations in the snow-on-ground fraction and snowfall fraction in milder climates, an increase in winter temperatures acts to increase the importance of the latter two while making the variability in total precipitation less important.

## 8. Conclusions

In the Introduction, three main questions were posed: (i) which factors control the interannual variability of snow amount in northern Europe, (ii) how does the dynamics of the interannual variability differ from that of the projected long-term climate change, and (iii) how does the long-term climate change affect the dynamics of interannual variability. The answers, based on the ERA5-Land reanalysis and the EURO-CORDEX RCM simulations, can be summarized as follows.

1. There is a contrast in the dynamics of interannual SWE variability between the colder (northern areas, Scandinavian mountains) and milder parts of northern Europe. In the former, variations in total precipitation dominate the SWE variability in most of the snow season. Together with a positive interannual correlation between winter temperature and precipitation, this leads to larger SWE in milder winters. In warmer areas, however, variations in SWE are mainly governed by variations in the snow-on-ground fraction (hence efficiency of snowmelt) and snowfall fraction. Therefore, there is less snow in milder winters.

2. Future changes in long-term mean SWE reflect a competition between increasing winter precipitation and reduced snowfall and snow-on-ground fractions. However, the latter two dominate practically everywhere in the area, leading to reduced SWE. The generally positive interannual correlation between SWE and temperature in the colder parts of northern Europe does not, therefore, correctly predict the long-term climate response. Still, in agreement

with the earlier ENSEMBLES RCM simulations (Räisänen and Eklund, 2012), the relative decrease in SWE is smaller in the colder than the milder parts of the domain.

3. Greenhouse gas induced warming affects the dynamics of interannual SWE variability in a manner analogous to the present-day geographical contrasts in this dynamics. Thus, in a warmer future climate, the relative impact of total precipitation on SWE variability tends to be reduced, whereas the variations in the snow-on-ground and snowfall fractions gain more importance.

This study relied on the ERA5-Land reanalysis in diagnosing the interannual SWE variability. The use of a reanalysis instead of direct observations was dictated by the lack of observations for the snowfall and snow-on-ground fractions (in-situ observations of SWE are also limited in number). The good agreement on snow depth between ERA5-Land and station observations (Fig. 3) is encouraging, suggesting that the dynamics of SWE variability may also be well represented. Still, the model-dependence of reanalysis products might affect some of the current results. For example, a good simulation of SWE might hide compensating errors in the snowfall fraction and snow-on-ground fraction, which are both difficult to verify but are potentially sensitive to the simulation of precipitation microphysics and the description of snowmelt, respectively. Unfortunately, few if any comparable data sets are currently available, since most reanalyses have coarser resolution than ERA5-Land and/or have artificial sources or sinks of snow due to the assimilation of snow observations (as, for example, in the parent ERA5 reanalysis). Regarding the simulation of the snow-on-ground fraction, off-line comparison of land surface models represents one way forward (Essery et al., 2020).

The big picture, in which interannual SWE variability is dominated by variations in winter precipitation in colder areas and by variations in the snow-on-ground and snowfall fractions in milder areas is, however, consistent with simple physical reasoning. On one hand, the winter total precipitation has a stronger effect on SWE where much of the precipitation falls as snow and most of the accumulated snowfall survives on ground; on the other hand the phase of precipitation and occurrence of melting episodes become increasingly sensitive to temperature variability where the mean temperature approaches zero. These considerations qualitatively explain both the geographical contrasts in the drivers of the present-day SWE variability and the shift towards increasingly snow-on-ground and snowfall fraction dominated SWE variability in a warmer future climate. Under a scenario with smaller greenhouse gas emissions, this shift as well as the changes in mean SWE would most likely proceed more slowly than the present results for RCP8.5 indicate, and it would take longer for them to rise above the background of natural variability. However, the qualitative similarity between the multi-RCM mean projections for 2059/60-2097/98 and the midway period 2020/21-2058/59 (Figs. 9, 10, 12 and 13) suggests that the basic characteristics of these changes should be largely insensitive to the magnitude of the radiative forcing.

A key message from this study is that interannual variability is, at best, an imperfect analogy for the effects of long-term climate change on snow conditions in northern Europe. We argue that this is because the relationship between the two main

atmospheric drivers of SWE variability, temperature and precipitation, differs between the interannual and climate change time scales. This difference most likely reflects the much larger role of atmospheric circulation in interannual variability (Saffioti et al., 2016; Räisänen, 2019) than in the forced greenhouse gas induced climate change, which is to a large extent driven by the radiative effect of increasing greenhouse gases and the resulting thermodynamic feedbacks (Collins et al., 2013).

**Appendix A**

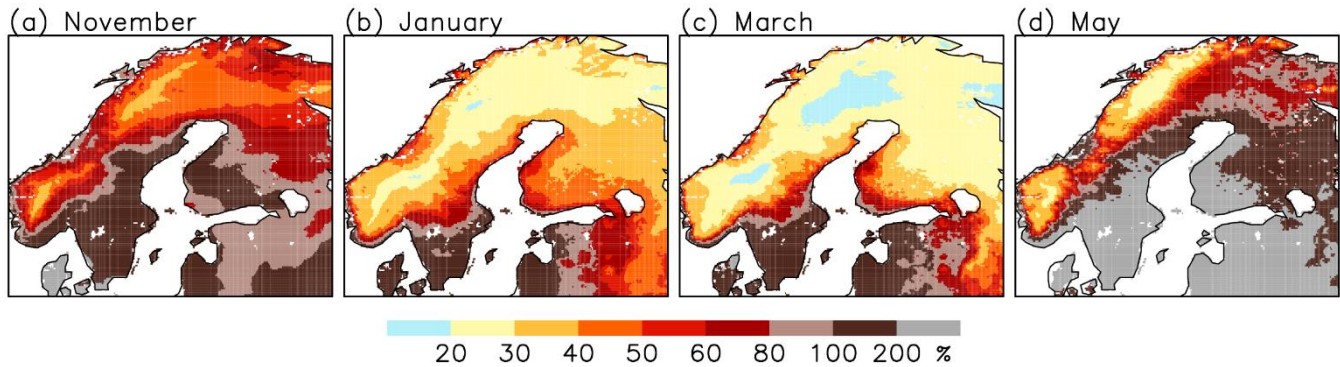

**Figure A1.** Coefficient of variation of monthly mean SWE in ERA5-Land in (a) November, (b) January, (c) March and (d) May.

**Data availability.** ERA5-Land data are available from the Copernicus Climate Change Service (C3S) Climate Date Store (https://cds.climate.copernicus.eu/cdsapp#!/search?type=dataset&text=era5-land), the EURO-CORDEX RCM simulations
from the Earth System Grid Federation (ESGF) (https://esgf-data.dkrz.de/search/cordex-dkrz/), and the station observations for Helsinki and Sodankylä from the Finnish Meteorological Institute (https://en.ilmatieteenlaitos.fi/download-observations).

**Author contribution.** JR designed the study, conducted the data analysis and wrote the manuscript.

**Competing interests.** The author declares that he has no conflict of interest.

**Acknowledgments**. The author thanks Adrià Fontrodona-Bach and two anonymous reviewers for their constructive comments.

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
