# Peer review of "Snow conditions in northern Europe: the dynamics of interannual variability versus projected long-term change"

_The Cryosphere, 2020_

## Referee Comment (RC1) · Anonymous Referee #1 · 16 Jan 2021

Summary;

The author discusses the mechanisms behind interannual variability in snow depth/SWE in Northern Europe, and how these mechanisms differ from that of the projected long-term climate change. Analysis is performed on the ERA5-land data set and regional climate models from EURO-CORDEX, by an earlier published method de-composing the contributions from change in total precipitation, snowfall fraction and snow-on-ground fraction (melting).

The paper is well written, easy to follow and fit well in The Cryosphere. The work is relevant for everyone that wants to understand the mechanisms behind changes in

snow on ground in more detail and explains why "warm winters" of today's climate can't be used as an analogy to future projected climate change with respect to snow cover. In general I like the paper very much, but have some comments and questions which hopefully will improve and clarify certain aspects.

General comments;

The scientific problem investigated is in itself interesting and line 59-60 is a start to put the work in a broader context and meaning. However, in the introduction I miss some further elaboration on why this work is important for a wider audience. Related to this, in the conclusions, is it possible to say something on the consequences of the findings in the study. For example (my understanding) does it mean that the accuracy of the snow schemes (i.e. melting processes) is important to get correct to simulate future changes in snow amounts in northern Europe?

The study refers to some earlier works, but not very many. However a google search shows many results on scientific papers on snow cover in Europe and also with examples of using data from, e.g. Sodankyla. This reviewer doesn't know the details of these papers, but imagine that at least some of them might be relevant to mention in the context of this work. For example the recent Cryosphere study looks relevant(?)

Essery, R., Kim, H., Wang, L., Bartlett, P., Boone, A., Brutel-Vuilmet, C., Burke, E., Cuntz, M., Decharme, B., Dutra, E., Fang, X., Gusev, Y., Hagemann, S., Haverd, V., Kontu, A., Krinner, G., Lafaysse, M., Lejeune, Y., Marke, T., Marks, D., Marty, C., Menard, C. B., Nasonova, O., Nitta, T., Pomeroy, J., Schädler, G., Semenov, V., Smirnova, T., Swenson, S., Turkov, D., Wever, N., and Yuan, H.: Snow cover duration trends observed at sites and predicted by multiple models, The Cryosphere, 14, 4687–4698, https://doi.org/10.5194/tc-14-4687-2020, 2020.

Even if it won't change the results from this study it would be nice to tie the present work more together with a broader part of earlier work.

[Figure]

The main work in this study relies heavily on the ERA5-Land data set. Some verification of ERA5-Land is also included at Sodankyla and Helsinki. The results of the verification in itself are encouraging. However, only comparing ERA5-Land with observations at two locations when discussion the results also in other terrain types, e.g. Scandinavian mountains are a bit scarce. I can imagine that the errors might be larger at other locations. If other studies of the quality of ERA5-land for northern Europe exist they could be referred to. I also miss some verification of the precipitation phase in ERA5-Land, it should be possible to compare from observations, either by manual observations (if available) or by using some temperature thresholds based on the observations. Finally for the verification part, is it possible and meaningful to estimate any of the decomposed contributions to snow variability from observations and compare them with ERA5-Land?

While the presented analysis relies on mean temperatures, how will the local temperature variability impact, e.g. in some areas there might be large variability with several "above zero degree" periods while in other areas it will be less variability and maybe constant temperatures under zero with the same mean temperature? I'm not suggesting any new analysis, but I'm curious about if you have any opinions on this subject? Can this partly explain why the correlation of Fig2a doesn't follow the mean temperatures?

Specific/minor comments;

Line 26: Since the example is one single winter (2019/20) it is not an example of large interannual variations in my view. Is it better to write something like: "An example of a particular anomalous winter was the winter 2019/20 . . . . . . ."?

Line 28-29: I suggest to add some details about the regional contrasts in the text, not only refer to Fig 1c. That would improve the flow in the reading.

Line 30-31: Add "(indicated by stippling in Fig 1c)" at the end of the sentence? I used some time to wonder how you knew that this was record low.

[Figure]

Line 42: Can the word "interannual" be skipped in this sentence?

Line 68-69: Since SWE and snow depth are not equal they are not necessarily straight-forward to compare, e.g. a change in snow depth can also come from changes in snow density. Is this important or discussed anywhere in the references?

Line 75: In my first read I wondered what the exact meaning of "the fraction of accu-mulated snowfall that remains on the ground" was. It is explained better later, but can something to clarify what it is be added already here (e.g. "the partition not melting")?

Line 102: Without having detailed knowledge about the local terrain/weather around Helsinki and Sodankyla I wonder if at least the coast line near Helsinki introduces some gradients in temperature? But as it is commented, since both Helsinki and Sodankyla observations have a long observation track and distribute observations via GTS they are probably assimilated in ERA5. I think that in combination with the fact that you com-pare monthly mean and not hourly values ensure the very high correspondence with observations? I guess the correspondence with observations are reduced somewhat if hourly values are compared.

Line 103-106: I agree with the reasoning and think that it is actually quite difficult to say if the true bias of ERA5 precipitation is positive or negative. Can you estimate how large, in percent, the apparent bias is? That can make it easier to judge (if you also know the gauge equipment characteristics and wind climate). I don't think it is necessary to be very quantitative, but it could strengthen your qualitative statement.

Figure 3: I think you can argue that the temperature bias is smaller during precipitation events. In particular the positive bias in Sodankyla probably arises from stably stratified situations with little precipitation.

Equation 2: It would be nice with some more details on how the different terms are calculated from ERA5-Land and the RCMs. For example, for the snowfall, do you use snowfall from ERA5-land (?) or do you use some temperature thresholds on the

total precipitation to decide precipitation phase? Are snowfall available from all RCMs? Even more, I'm curious on how you calculate the G-term in practise. Can you give some more details?

Line 150-151: Isn't "1" applied on single winters and then the average is taken over all these winters? Could the word "decomposed" be added in front of "values for an individual winter"?

Line 152-153: Again isn't "1" applied on single winters and then averaged? Or do I misunderstand?

Equation 3 & 4: I don't understand these. I think they need some more explanations. That would help me to understand e.g. Line 206-207 also.

Figure 7: What does the "C" in "SDC" stand for?

Line 261: add "change" after "SWE"?

Figure 10. The discussion of Fig 10 is not clear to me. I don't fully understand what the message is. Can this be made clearer?

Line 426: "colder areas" Is it possible based on this study to quantify/define what "colder areas" are? Wouldn't that be of interest for many, when a local climate enters a new regime where variations in winter precipitation is no longer the main driver?

---

## Referee Comment (RC2) · Anonymous Referee #2 · 2 Feb 2021

Summary

This is an interesting and illustrative study documenting the impacts of different drivers on snow cover variability in northern Europe and evaluating the impact of global warming on these drivers. Specifically, the study elaborates why rising temperatures with increasing precipitation levels do not result in increasing snow depth even in the coldest areas of the study region where milder winters tend to be more snowier than colder winters. The study is in general well written and easy to follow and I have only a few comments as outlined below.

Specific comments:

[Figure]

Part of the cited literature is not included in the reference list, e.g., Lehtonen, 2015 and Räisänen, 2019.

The two figures in the Introduction section fit very well in the section.

Lines 87-93: I agree with the editor that it would be best to introduce whether ERA5-Land assimilates in-situ measurements and/or remotely sensed data or not in the first paragraph of this section.

Lines 110 and 443: The link to the FMI website can be given in English as follows: https://en.ilmatieteenlaitos.fi/download-observations

Line 143: Is the non-linear term included into the equation just to explain residual SWE variations not explained by the first three terms?

Fig. 3: Just as a side note, perhaps the negative temperature bias in ERA5-Land in Helsinki is at least partly due to urban heat island effect whereas in Sodankylä the positive bias is likely most significant in cold weather situations with marked temperature inversion. It even seems that the bias is larger in cold than mild winters, supporting this latter hypothesis.

Fig. 7: I am just wondering whether it would be more interesting to show in the second panel the ratio of standard deviation of SWE to the mean SWE, i.e., SD(SWE)/Mean(SWE)? I am not saying it would be a better option but left the choice for the author, but this would highlight more clearly the point raised on lines 233-234 about the areas with higher variability. Perhaps it would just mirror the left panel.

Fig. 8: I'll suggest a small change in the caption. It would be clearer to state that "Middle: the changes from 1981/82–2019/20 to 2020/21–2058/59..." and "Right: as middle, but from 1981/82–2019/20 to 2059/60–2097/98"

Fig. 10: Interestingly, the largest precipitation increase in relative terms is projected to take place in the eastern side of the Scandinavian mountains (Fig. 10b), although increase in the easterlies is not usually projected by climate models in this region.

Comparison of Fig 10b with Figs 10d and 10e even suggests a shift towards dynami-cally colder weather types as the projected increase in precipitation is smallest (virtu-ally negligible) along the Norwegian coast, where the positive coupling between mean temperature and precipitation is the strongest, while the only areas in Sweden where precipitation levels tend to be even slightly higher in cold than warm years are among the areas with largest projected increase in precipitation.

Fig. 11: It seems like the model data has a warm bias at least in Sodankylä, as the interannual standard deviation of SWE peaks already in April indicating an earlier melt season compared to ERA5-Land showing the peak in May. Perhaps it is thus unlikely that even in the case of RCP8.5, the peak melt season would shift to as early as March by the end of century.

Lines 418-423: Nice that the sources of uncertainty are recognized and acknowledged by the author. This is not obvious in all the studies.

Lines 433-439: I think this is a rather expected conclusion. It would be surprising if a winter with so extremely extraordinary circulation patterns than 2019/20 would be an analogous winter of the future as the winters dynamically analogous to 2019/20 will remain rare in the future, though potentially turning slightly more common than in the past.

---

## Referee Comment (RC3) · Adrià Fontrodona-Bach (Referee) · 8 Feb 2021

**General comments:**

This interesting manuscript provides an in-depth analysis of interannual SWE variability and long-term climate change effects on SWE over northern Europe. The study uses reanalysis data and Regional Climate Models under climate change scenario RCP8.5. The author disentangles the components of SWE variability into the contribution from three components. This provides a clear view on the effects and interplay between warming temperatures and increasing precipitation over northern latitudes. Here, a clear distinction is made between (i) the effect of rising temperature and precipitation due to climate change, showing that temperature clearly dominates the future climate leading to an overall decrease in SWE, and (ii) the fact that in the current climate, warmer years have higher SWE due to different prevailing atmospheric conditions leading to higher precipitation (when still cold enough). Although there is some uncertainty associated to the methods used, the results are robust, novel, and provide a great contribution to scientific knowledge on the effect of climate change on snow. The manuscript is well structured and well written, so I support its publication. I can only add a few comments to clarify and generate discussion on a couple of matters.

**Specific Comments**

- Line 57: What is the reason that such a low correlation (r>0.32) is significant at 5% level? Very high variability?

- Lines 59-69: Although I agree that only "the first part of the reasoning is correct" regarding the analogy with the future climate, there is also observation-based research showing that snowfall and snow depth have already been increasing over some parts of Scandinavia and Eurasia (even if the reasons are not entirely clear). I suggest to extend this paragraph and discuss these references too:

- https://doi.org/10.2166/nh.2012.109

- https://doi.org/10.1029/2018GL079799

- https://doi.org/10.5194/tc-12-227-2018

- March is chosen as a key month because of its maximum in SWE over most of the area, but March SWE is dependent on P and T of the previous winter months too. In Figures 1,2,3 March SWE is compared to variability in NDJFM temperature and precipitation. However, in Figures 5, 9 and 12, it is not clear to me if the decomposition of variability into the three components is done only for March or for the entire winter.

Perhaps this is clear from the mathematical theory presented, but a clarification and justification of this would be appreciated.

- Table 2: Might be my lack of understanding, but I do not know what the values in parentheses mean. What are the "individual terms"? How are they different from the four rhs terms in Eq. 2? It would be helpful to clarify this.

- Lines 333-335. Regarding the positive correlation between SWE variability and temperature due to the westerly flow anomalies. Would the RCMs considered here, with boundary conditions from GCMs, reproduce any change in these anomalies which could have a strong effect in the future? Maybe just worth discussing this possibility.

- Lines 354-356: Could this relate to the contrasting response of mean snowfall and extreme snowfall to warming as shown in https://doi.org/10.1038/nature13625 and https://doi.org/10.1007/s00382-015-2587-0

- Given the choice of scenario RCP8.5, and the sensitivity of this type of research to crossing or not crossing the freezing threshold (snow or no snow), it would be good to raise a point in the conclusions whether how different might the results under another scenario. Or to call for future work on the analysis of multiple scenarios.

**Technical corrections**

- Line 57 in caption should be: (c) NDJFM mean precipitation (not temperature)

- Lines 150-154: Please add also "Equation" to "1" and "2", to clarify.

- Line 319: Should be Fig. 10c (not 9c).

---

## Author Comment (AC1) · 20 Feb 2021

**Response to Reviewer 1**

I thank the reviewer for his / her valuable comments on the manuscript. My response to the comments and the changes I plan to make in the revised manuscript are detailed below. For clarity, the comments are in blue font, while my response is in black. In some cases, I have included text planned to appear in the revised manuscript in red font.

The author discusses the mechanisms behind interannual variability in snow depth / SWE in Northern Europe, and how these mechanisms differ from that of the projected long-term climate change. Analysis is performed on the ERA5-land data set and regional climate models from EURO-CORDEX, by an earlier published method decomposing the contributions from change in total precipitation, snowfall fraction and snow-on-ground fraction (melting). The paper is well written, easy to follow and

10 fit well in The Cryosphere. The work is relevant for everyone that wants to understand the mechanisms behind changes inC1snow on ground in more detail and explains why "warm winters" of today's climate can't be used as an analogy to future projected climate change with respect to snow cover. In general I like the paper very much, but have some comments and questions which hopefully will improve and clarify certain aspects

**General comments**

15 The scientific problem investigated is in itself interesting and line 59-60 is a start to put the work in a broader context and meaning. However, in the introduction I miss some further elaboration on why this work is important for a wider audience. Related to this, in the conclusions, is it possible to say something on the consequences of the findings in the study. For example (my understanding) does it mean that the accuracy of the snow schemes (i.e. melting processes) is important to get correct to simulate future changes in snow amounts in northern Europe?

20

This is a good point. I plan to add the following paragraph towards the end of the Introduction:

The significance of this research in a wider perspective is twofold. First, a better understanding of the processes involved in the interannual variability and long-term trends of snow conditions is valuable for model developers, helping to focus the

25 development work towards the most important processes. For example, the findings in this paper suggest that, in areas with relative mild winters like southern Finland, the description of snowmelt is important for the simulation of both the interannual variability and future trends of snow amount. Second, the current results bear an important message for climate impact researchers and the general audience, by showing why the snow conditions in individual mild winters are not a perfect analogy for what to expect in the future.

30

The study refers to some earlier works, but not very many.

However a google search shows many results on scientific papers on snow cover in Europe and also with examples of using
data from, e.g. Sodankyla. This reviewer doesn't know the details of these papers, but imagine that at least some of them might
be relevant to mention in the context of this work. For example the recent Cryosphere study looks relevant (?)

Essery, R., Kim, H., Wang, L., Bartlett, P., Boone, A., Brutel-Vuilmet, C., Burke, E.,Cuntz, M., Decharme, B., Dutra, E., Fang,
X., Gusev, Y., Hagemann, S., Haverd,V., Kontu, A., Krinner, G., Lafaysse, M., Lejeune, Y., Marke, T., Marks, D., Marty,C.,
Menard, C. B., Nasonova, O., Nitta, T., Pomeroy, J., Schädler, G., Semenov, V.,Smirnova, T., Swenson, S., Turkov, D., Wever,
N., and Yuan, H.: Snow cover duration trends observed at sites and predicted by multiple models, The Cryosphere, 14,4687–
4698, https://doi.org/10.5194/tc-14-4687-2020, 2020.Even if it won't change the results from this study it would be nice to tie
the presentwork more together with a broader part of earlier work.

Thanks for this reference. The mentioned study nicely exemplifies how off-line comparison of land surface models could be
used to study the uncertainty in such processes (e.g., snow melt) for which direct observations are not available. I plan to
include in the concluding section of the revised manuscript.

Considering the comments of all three reviewers and my own literature search, I also plan to add 10-15 other new references
in the revised manuscript.

The main work in this study relies heavily on the ERA5-Land data set. Some verification of ERA5-Land is also included at
Sodankyla and Helsinki. The results of theverification in itself are encouraging. However, only comparing ERA5-Land with
observations at two locations when discussion the results also in other terrain types, e.g. Scandinavian mountains are a bit
scarce. I can imagine that the errors might be larger at other locations. If other studies of the quality of ERA5-land for northern
Europe exist they could be referred to.

ERA5-Land is still a new data set, and therefore few verification studies are available. I have found a couple of examples that
I may mention in the revised manuscript, but I am not aware of anything focussing on northern Europe.

Also, it is true that the verification provided in the manuscript is far from exhaustive. This will be clearly acknowledged in the
revised manuscript, where I will also discuss some further aspects that could (or at least ideally should) be addressed in further
work, approximately as follows (after Fig. 3 in Section 2):

65 The comparison presented in Fig. 3 is far from exhaustive. More insight could be gained, for example, by extending the evaluation to the daily time scale, but this is out of the focus of the present study. Another unverified aspect is the ability of ERA5-Land to distinguish between solid and liquid precipitation in near-zero temperatures. This is important because, in principle, a good simulation of snow amount might still hide compensating errors in snowfall and snowmelt. Unfortunately, there is no ground truth to compare with, since precipitation measurements in Finland only record the total precipitation.

70 Empirical estimates for the dependence of the snowfall/rainfall probability on near-surface temperature and humidity have been derived based on synoptic observations (e.g., Auer, 1974; Koistinen et al., 2004), but the conversion to total daily snowfall or rainfall fractions is nontrivial because precipitation intensity, temperature and humidity all vary on sub-daily time scales.

I also miss some verification of the precipitation phase in ERA5-Land, it should be possible to compare from observations,

75 either by manual observations (if available) or by using some temperature thresholds based on the observations.

I agree that the ability of ERA5-Land to represent the phase of precipitation is one of the main uncertainties in the study. The problem is that verification of this is far from trivial. The visually or optically determined phase of precipitation is encoded in the present-weather code in SYNOP observations, but in practice these data are not easily available, would require substantial

80 work to process, the processing would require subjective choices (e.g. how to deal with sleet?), and the conversion to precipitation totals would require information on precipitation intensity on sub-daily time scales. Approximate temperature-based methods (as mentioned in the previous red paragraph) might be slightly more straightforward to use, but still represent an approximation and would require data on temperature and precipitation intensity on sub-daily time scales. Considering these issues, I feel it is better to leave this as a topic for further research. However, I will acknowledge this issue in the revised

85 manuscript (as indicated in the paragraph in red above).

Finally for the verification part, is it possible and meaningful to estimate any of the decomposed contributions to snow variability from observations and compare them with ERA5-Land?

90 This would certainly be useful, but is far from trivial in practice. The problem is the lack of direct observations for the snowfall fraction of total precipitation (as discussed above), together with the fact that measurements of SWE are scattered in space and available at irregular time intervals.

While the presented analysis relies on mean temperatures, how will the local temperature variability impact, e.g. in some areas

95 there might be large variability with several "above zero degree" periods while in other areas it will be less variability and may beconstant temperatures under zero with the same mean temperature? I'm not suggesting any new analysis, but I'm curious about if you have any opinions on this subject?

Can this partly explain why the correlation of Fig2a doesn't follow the mean temperatures?

100

Differences in temperature variability may well play a role, but it is difficult to separate this from other factors. Among them, differences in total winter precipitation (i.e. much more precipitation falling in western Norway than elsewhere in the study area) appear to play an important role. To demonstrate this, and also to answer your last comment on what is meant by "colder areas", I plan to add the following new figure to the revised manuscript:

105

[Figure]

**Figure 8.** (a) The relative contribution of precipitation anomalies to the standard deviation of SWE in March as a function of the NDJFM mean temperature in 1981/82 − 2019/20. Each dot represents a single $0.1° \times 0.1°$ grid cell, coloured according to the mean NDJFM precipitation shown in (b). The solid line in (a) indicates the mean values for 1°C temperature bins, and the two dashed lines the mean ± one standard deviation.

110

together with the text in the next two paragraphs (the first one devoted for quantifying the temperature dependence and the second discussing the role of total precipitation):

115  As a further illustration, the relative contribution of precipitation variability on SWE variability in March (row 3, column 3 in Fig. 7) is plotted as a function of the climatological NDJFM mean temperature in Fig. 8a. On the average, this contribution exceeds 80% where $T_{NDJFM} < -11°C$, is close to 50% where $T_{NDJFM} \approx -7°C$, and decreases to zero at $T_{NDJFM} \approx -2°C$. Despite the non-linearity of the relationship, there is a strong negative spatial correlation ($r = -0.85$) between the two variables

in Fig. 8a. Conversely, the relative contributions of snowfall fraction variability ($sdc(\Delta F)/sdc(SWE)$) and snow-on-ground fraction variability ($sdc(\Delta G)/sd(SWE)$) are positively correlated with the NDJFM mean temperature ($r = 0.65$ and $0.83$, respectively).

Nevertheless, the dynamics of interannual SWE variability is not solely controlled by the winter mean temperature. For the same NDJFM mean temperature, and excluding the mildest areas, $sdc(\Delta P)/sd(SWE)$ tends to increase with increasing NDJFM mean precipitation (see the colour coding in Fig. 8a). In particular, the SWE variability in western Norway, where more precipitation falls than elsewhere in Northern Europe (Fig. 8b), is more strongly affected by precipitation variability than expected from the winter mean temperature alone. On one hand, the larger mean precipitation is associated with larger absolute precipitation variability. On the other hand, larger amounts of snowfall reduce the variability in the snow-on-ground fraction, because a larger amount of snowmelt is needed for a unit change in the latter.

**Specific/minor comments;**

Line 26: Since the example is one single winter (2019/20) it is not an example of largeinterannual variations in my view. Is it better to write something like: "An example of a particular anomalous winter was the winter 2019/20 . . . . . . ."?

Will be changed approximately as suggested.

Line 28-29: I suggest to add some details about the regional contrasts in the text, notonly refer to Fig 1c. That would improve the flow in the reading.

I will add something like this: Record-breaking positive anomalies of 3-5°C in the November-to-March mean temperature extended from southern Sweden to southern and central Finland, the Baltic States and western Russia, whereas the precipitation surplus was unusually large especially in Finland.

Line 30-31: Add "(indicated by stippling in Fig 1c)" at the end of the sentence? I used some time to wonder how you knew that this was record low.

Will be done.

C3Line 42: Can the word "interannual" be skipped in this sentence?

Yes, I will skip it.

Line 68-69: Since SWE and snow depth are not equal they are not necessarily straightforward to compare, e.g. a change in snow depth can also come from changes in snow density. Is this important or discussed anywhere in the references?

I will add a brief note ("Note, though, that snow depth is affected by snow density as well as SWE") on this in the text.

Line 75: In my first read I wondered what the exact meaning of "the fraction of accumulated snowfall that remains on the ground" was. It is explained better later, but can something to clarify what it is be added already here (e.g. "the partition not melting")?

I will add a brief explanation as: the fraction of accumulated snowfall that has not yet melted and thus remains on ground (snow-on-ground fraction)

Line 102: Without having detailed knowledge about the local terrain/weather around Helsinki and Sodankyla I wonder if at least the coast line near Helsinki introduces somegradients in temperature? But as it is commented, since both Helsinki and Sodankylaobservations have a long observation track and distribute observations via GTS theyare probably assimilated in ERA5. I think that in combination with the fact that you com-pare monthly mean and not hourly values ensure the very high correspondence with observations? I guess the correspondence with observations are reduced somewhat ifhourly values are compared.

It is indeed possible, and even likely, that the specifics of the local station environments explain some of the apparent biases in ERA5-Land. It also seems likely that the correspondence on daily or hourly time scales would be worse than that for monthly or winter mean values. I will mention these issues briefly when discussing Figure 3 in the revised manuscript, but only briefly as these details are somewhat out of the focus of the study.

Line 103-106: I agree with the reasoning and think that it is actually quite difficult to say if the true bias of ERA5 precipitation is positive or negative. Can you estimate how large, in percent, the apparent bias is? That can make it easier to judge (if you also know the gauge equipment characteristics and wind climate). I don't think it is necessary to be very quantitative, but it could strengthen your qualitative statement.

Thanks for this question. A closer look at Taskinen and Söderholm (2016, full reference in the original manuscript) allows the following addition at this point: "The absolute mean values in the reanalysis exceed the station measurements by 12% in Helsinki and 18% in Sodankylä, … In fact, the difference between ERA5-Land and the station observations agrees well with

185 Taskinen and Söderholm (2016), who estimate the average December-to-March precipitation in Finland and its cross-boundary watersheds in 1982-2011 to have been 17.5% larger than measured".

Figure 3: I think you can argue that the temperature bias is smaller during precipitation events. In particular the positive bias in Sodankyla probably arises from stably stratified situations with little precipitation.

190

This is plausible. However, to keep the focus of the paper, I probably won't comment on this in the revised manuscript – in particular as the cold winter climate in Sodankylä means that snow amount is more strongly controlled by precipitation than temperature.

195 Equation 2: It would be nice with some more details on how the different terms arecalculated from ERA5-Land and the RCMs. For example, for the snowfall, do you use snowfall from ERA5-land (?) or do you use some temperature thresholds on theC4 total precipitation to decide precipitation phase?

To clarify, this addition will be included to the paragraph following Eq. (1): "All the variables required in Eq. (1) (i.e., the total
200 precipitation, snowfall and SWE) are directly available for both ERA5-Land and the EURO-CORDEX simulations."

Are snowfall available from all RCMs? Even more, I'm curious on how you calculate the G-term in practise. Can you give some more details?

205 Yes, snowfall is available for all the RCMs included in this study (see above). The snow-on-ground fraction $G$ is obtained by dividing SWE by the accumulated (i.e. time integrated) snowfall. I will add a brief explanation on this below Eq. (1) in the revised manuscript.

Line 150-151: Isn't "1" applied on single winters and then the average is taken overall these winters? Could the word
210 "decomposed" be added in front of "values for anindividual winter"?

Sorry for being unclear with the notations. "1" and "2" here referred to the subscripts introduced just above Eq. (1). The planned clarification for this is as follows:

215 In this study, the decomposition (2) is applied in two different ways:

1. When studying interannual variations in SWE, $X_1$ as defined above Eq. (2) represent the mean values for a 39-winter period (1981/82 to 2019/20, 2020/21 to 2058/59 or 2059/60 to 2097/98) and $X_2$ the values for an individual winter.

2. When studying long-term changes in SWE, $X_1$ represent the mean values for winters 1981/82 to 2019/20, and $X_2$ those for either 2020/21 to 2058/59 or 2059/60 to 2097/98.

Line 152-153: Again isn't "1" applied on single winters and then averaged? Or do I misunderstand?

See the clarification above. Your misunderstanding was entirely my fault!

Equation 3 & 4: I don't understand these. I think they need some more explanations.That would help me to understand e.g. Line 206-207 also.Figure 7: What does the "C" in "SDC" stand for?

SDC stands for "standard deviation contribution". I will add a note on this. Also, to make the statistics easier to follow, I will provide a longer derivation in the revised manuscript, as follows:

Multiplying Eq. (2) with $\Delta SWE$ and averaging over a 39-winter period, the interannual variance of SWE can be decomposed to the contributions of the four right-hand-side (rhs) terms in Eq. (2) as

$$var(SWE) = \langle \Delta SWE^2 \rangle = \langle \Delta SWE \sum_{i=1}^{4} \Delta SWE_i \rangle = \sum_{i=1}^{4} cov(\Delta SWE_i, SWE) \tag{3}$$

where the angle brackets indicate a time mean, $var$ is variance and $cov$ covariance. Similarly, the standard deviation ($s$) of SWE is decomposed as

$$s(SWE) = \frac{var(SWE)}{s(SWE)} = \sum_{i=1}^{4} \frac{cov(\Delta SWE_i, SWE)}{s(SWE)} = \sum_{i=1}^{4} \frac{cov(\Delta SWE_i, SWE)}{s(SWE)s(\Delta SWE_i)} s(\Delta SWE_i) \tag{4}$$

which can be rewritten using the definition of correlation ($r$) as

$$s(SWE) = \sum_{i=1}^{4} sdc_i = \sum_{i=1}^{4} r(\Delta SWE_i, SWE)s(\Delta SWE_i) \tag{5}$$

where the $sdc_i$:s refer to the *standard deviation contributions* of the four rhs terms in Eq. (2).

Line 261: add "change" after "SWE"?

Suggestion accepted.

Figure 10. The discussion of Fig 10 is not clear to me. I don't fully understand what the message is. Can this be made clearer?

The main point in Fig. 10 (to be Fig. 11 in the revised manuscript) is the difference between the c- and d-panels, which shows that, for each 1°C of warming, the projected long-term increase in precipitation is different, and generally smaller, than the precipitation anomalies accompanying a 1°C interannual winter temperature anomaly in ERA5-Land. Here is the planned revision, also taking into account the comments of the other reviewers:

In apparent conflict with the simulated future decrease in SWE nearly everywhere in northern Europe, Fig. 2a showed a positive interannual correlation between March mean SWE and NDJFM mean temperature over the Scandinavian mountains and in the northern parts of Sweden and Finland. This conflict arises because the relationship between winter temperature and precipitation differs between the long-term climate change and the interannual variability. As discussed below based on Fig. 11, the projected long-term increase in winter precipitation is in most of northern Europe smaller than the projected warming together with the interannual regression relationship between temperature and precipitation anomalies in ERA5-Land would suggest.

The EURO-CORDEX RCMs simulate, on the average, a NDJFM mean warming of ca. 3-5 °C from 1981/82-2019/20 to 2059/60-2097/98, with a general increase from southwest to northeast (Fig. 11a). The change in precipitation varies from slight local decreases in western and northern Norway to increases of up to 25 %, with a relatively sharp northwest-to-southeast contrast across the Scandinavian mountains (Fig. 11b). This contrast is qualitatively similar to that found by Räisänen and Eklund (2012), but its connection to the atmospheric circulation in the EURO-CORDEX RCMs would require further investigation. The multi-RCM mean changes in the NDJFM mean sea level pressure in northern Europe are small (from 0 to +1 hPa), implying only very modest changes in the average lower tropospheric winds (not shown).

The ratio between the precipitation and temperature changes is mostly 2-6 % $(°C)^{-1}$, but lower in western and northern Norway (Fig. 11c). On the interannual time scale, however, a 1 °C positive temperature anomaly is statistically accompanied by a 12-15 % precipitation anomaly in western Norway (Fig. 11d), where westerly flow anomalies result both in advection of warm Atlantic air and forced ascent uphill the Scandinavian mountains. The interannual regression coefficient (Fig. 11d) also exceeds the long-term precipitation-to-temperature change ratio (Fig. 11c) in Finland and northern Sweden. For example, in the grid box closest to Sodankylä, the long-term change ratio (3.4 % $(°C)^{-1}$) is only half of the interannual slope (6.1 % $(°C)^{-1}$) in ERA5-Land. The interannual regression coefficients in the EURO-CORDEX RCMs agree generally well with ERA5-Land (not shown).

As indicated by the last sentence, I plan to omit the original Fig. 10e, because this panel was not central to the main message.

Line 426: "colder areas" Is it possible based on this study to quantify/define what "colder areas" are? Wouldn't that be of interest for many, when a local climate enters a new regime where variations in winter precipitation is no longer the main driver?

285

This comment is addressed in the new Figure 8 (and related text) that was included earlier in this response letter.

---

## Author Comment (AC2) · 20 Feb 2021

**Response to Reviewer 2**

I thank the reviewer for his / her valuable comments on the manuscript. My response to the comments and the changes I plan to make in the revised manuscript are detailed below. For clarity, the comments are in blue font, while my response is in black. In some cases, I have included text planned to appear in the revised manuscript in red font.

5 **Summary**

This is an interesting and illustrative study documenting the impacts of different drivers on snow cover variability in northern Europe and evaluating the impact of global warming on these drivers. Specifically, the study elaborates why rising temperatures with increasing precipitation levels do not result in increasing snow depth even in the coldest areas of the study region where milder winters tend to be more snowier than colder winters. The study is in general well written and easy to
10 follow and I have only a few comments as outlined below.

**Specific comments:**

Part of the cited literature is not included in the reference list, e.g., Lehtonen, 2015 and Räisänen, 2019.

It is a shame to confess that I apparently forgot to check the list of references. In addition to Lehtonen (2015) and Räisänen
15 (2019), van Vuuren et al. (2011) was also missing. The details of these three references are as follows:

Lehtonen, I., Four consecutive snow-rich winters in Southern Finland: 2009/2010–2012/2013. Weather, 70, 3-8, doi: 10.1002/wea.2360, 2015.
Räisänen, J.: Effect of atmospheric circulation on recent temperature changes in Finland, Clim. Dyn., 53, 5675-5687, doi:
20 10.1007/s00382-019-04890-2, 2019.
van Vuuren, D. P., Edmonds, J., Kainuma, M., Riahi, K., Thomson, A., Hibbard, K., Hurtt, G. C., Kram, T., Krey, V., Lamarque, J.-F., Masui, T., Meinshausen, M., Nakicenovic, N., Smith, S. J., and Rose, S. K.: The representative concentration pathways: an overview. Climatic Change, 109, 5-31, doi:10.1007/s10584-011-0148-z, 2011.

25 Lines 87-93: I agree with the editor that it would be best to introduce whether ERA5-Land assimilates in-situ measurements and/or remotely sensed data or not in the first paragraph of this section.

As was mentioned in the original manuscript (L91), ERA5-Land uses no data assimilation (although, of course, the "parent" ERA5 renalysis does, and this affects the meteorological forcing seen by the H-Tessel land surface model used for generating
30 ERA5-Land). However, looking more closely at the documentation of ERA5, I found that I had been unclear about one detail:

although ERA5 assimilates observations of surface air temperature, it only assimilates precipitation measurements (to some extent) in North America, and not in Europe. This will be pointed out in the revised manuscript.

Lines 110 and 443: The link to the FMI website can be given in English as follows:
https://en.ilmatieteenlaitos.fi/download-observations

Thanks. This address will be changed in both places.

Line 143: Is the non-linear term included into the equation just to explain residual SWE variations not explained by the first three terms?

Yes, the inclusion of the non-linear term makes the equation exact (the derivation is simple, although not shown in the manuscript). I therefore retain this term, although it has little practical significance.

Fig. 3: Just as a side note, perhaps the negative temperature bias in ERA5-Land in Helsinki is at least partly due to urban heat island effect whereas in Sodankylä the positive bias is likely most significant in cold weather situations with marked temperature inversion. It even seems that the bias is larger in cold than mild winters, supporting this latter hypothesis.

Both of these speculations are probably correct. I plan to include a brief mention on the urban heat island effect. However, it is probably not necessary to discuss the inversions in Sodankylä, because this would require a longer explanation which could distract the flow of the text – particularly recalling that the cold winter climate in Sodankylä makes SWE generally less sensitive to temperature than precipitation.

Fig. 7: I am just wondering whether it would be more interesting to show in the second panel the ratio of standard deviation of SWE to the mean SWE, i.e., SD(SWE)/Mean(SWE)? I am not saying it would be a better option but left the choice for the author, but this would highlight more clearly the point raised on lines 233-234 about the areas with higher variability. Perhaps it would just mirror the left panel.

After considering this comment, I decided to retain Fig. 7 as is, to keep its interpretation as simple as possible. However, I have prepared separate maps for the coefficient of variation, which I plan to include as an appendix (see below):

[Figure]

**Figure A1.** Coefficient of variation of monthly mean SWE in ERA5-Land in (a) November, (b) January, (c) March and (d) May.

Fig. 8: I'll suggest a small change in the caption. It would be clearer to state that "Middle: the changes from 1981/82–2019/20 to 2020/21–2058/59: : :" and "Right: as middle, but from 1981/82–2019/20 to 2059/60–2097/98"

Suggestion accepted.

Fig. 10: Interestingly, the largest precipitation increase in relative terms is projected to take place in the eastern side of the Scandinavian mountains (Fig. 10b), although increase in the easterlies is not usually projected by climate models in this region. Comparison of Fig 10b with Figs 10d and 10e even suggests a shift towards dynamically colder weather types as the projected increase in precipitation is smallest (virtually negligible) along the Norwegian coast, where the positive coupling between mean temperature and precipitation is the strongest, while the only areas in Sweden where precipitation levels tend to be even slightly higher in cold than warm years are among the areas with largest projected increase in precipitation.

I also find this distribution of precipitation changes somewhat surprising, although it is qualitatively similar to that found by Räisänen and Eklund (2012, Climate Dynamics, 38, 2575-2591) for the earlier ENSEMBLES RCM simulations. I tried to study its origin by looking at the time mean sea level pressure changes in the EURO-CORDEX RCMs, but found these changes to be too small to provide any obvious explanation. This is thus clearly an issue for further research. The planned text discussing this in the revised manuscript is as follows:

The EURO-CORDEX RCMs simulate, on the average, a NDJFM mean warming of ca. 3-5°C from 1981/82-2019/20 to 2059/60-2097/98, with a general increase from southwest to northeast (Fig. 11a). The change in precipitation varies from slight local decreases in western and northern Norway to increases of up to 25%, with a relatively sharp northwest-to-southeast contrast across the Scandinavian mountains (Fig. 11b). This contrast is qualitatively similar to that found by Räisänen and Eklund (2012), but its connection to the atmospheric circulation in the EURO-CORDEX RCMs would require further

investigation. The multi-RCM mean changes in the NDJFM mean sea level pressure in northern Europe are small (from 0 to +1 hPa), implying only very modest changes in the average lower tropospheric winds (not shown).

Fig. 11: It seems like the model data has a warm bias at least in Sodankylä, as the interannual standard deviation of SWE peaks already in April indicating an earlier melt season compared to ERA5-Land showing the peak in May. Perhaps it is thus unlikely that even in the case of RCP8.5, the peak melt season would shift to as early as March by the end of century.

The early snowmelt during the baseline period naturally affects the quantitative interpretation of the future projections. This will be pointed out explicitly in the revised manuscript, as shown below. Note that I plan to add a new Figure 8 in the article (see the response to Reviewer 1), and the old Fig. 8 will therefore become Fig. 9 and Fig. 11 will become Fig. 12.

Note, though, that the standard deviation of SWE in Sodankylä in years 1982-2020 reaches its maximum earlier in the RCMs than in ERA5-Land (bottom left of Fig. 12 vs. Fig. 6b), just as the mean SWE does (bottom left of Fig. 9 vs. Fig. 4l). This bias naturally affects the quantitative interpretation of the model projections.

Lines 418-423: Nice that the sources of uncertainty are recognized and acknowledged by the author. This is not obvious in all the studies.

Thanks for this fcomment. In fact, considering the comments of the other reviewers, I decided to expand the discussion of uncertainties in the Conclusions section, as follows.

This study relied on the ERA5-Land reanalysis in diagnosing the interannual SWE variability. The use of a reanalysis instead of direct observations was dictated by the lack of observations for the snowfall and snow-on-ground fractions (in-situ observations of SWE are also limited in number). The good agreement on snow depth between ERA5-Land and station observations (Fig. 3) is encouraging, suggesting that the dynamics of SWE variability may also be well represented. Still, the model-dependence of reanalysis products might affect some of the current results. For example, a good simulation of SWE might hide compensating errors in the snowfall fraction and snow-on-ground fraction, which are both difficult to verify but are potentially sensitive to the simulation of precipitation microphysics and the description of snowmelt, respectively. Unfortunately, few if any comparable data sets are currently available, since most reanalyses have coarser resolution than ERA5-Land and/or have artificial sources or sinks of snow due to the assimilation of snow observations (as, for example, in the parent ERA5 reanalysis). Regarding the simulation of the snow-on-ground fraction, off-line comparison of land surface models represents one way forward (Essery et al., 2020).

120 Lines 433-439: I think this is a rather expected conclusion. It would be surprising if a winter with so extremely extraordinary circulation patterns than 2019/20 would be an analogous winter of the future as the winters dynamically analogous to 2019/20 will remain rare in the future, though potentially turning slightly more common than in the past.

I agree that this conclusion is not unsurprising, but I still hope and believe that it was worthwhile to demonstrate why
125 interannual variability is not a good analogy for the long-term climate change effect on snow conditions.

---

## Author Comment (AC3) · 20 Feb 2021

**Response to Reviewer 3**

Dear Dr. Fontana Bach,

I thank you for your valuable comments on the manuscript. My response to the comments and the changes I plan to make in the revised manuscript are detailed below. For clarity, the comments are in blue font, while my response is in black. In some
cases, I have included text planned to appear in the revised manuscript in red font.

**General comments:**

This interesting manuscript provides an in-depth analysis of interannual SWE variability and long-term climate change effects on SWE over northern Europe. The study uses reanalysis data and Regional Climate Models under climate change scenario RCP8.5. The author disentangles the components of SWE variability into the contribution from three components. This
provides a clear view on the effects and interplay between warming temperatures and increasing precipitation over northern latitudes. Here, a clear distinction is made between (i) the effect of rising temperature and precipitation due to climate change, showing that temperature clearly dominates the future climate leading to an overall decrease in SWE, and (ii) the fact that in the current climate, warmer years have higher SWE due to different prevailing atmospheric conditions leading to higher precipitation (when still cold enough). Although there is some uncertainty associated to the methods used, the results are
robust, novel, and provide a great contribution to scientific knowledge on the effect of climate change on snow. The manuscript is well structured and well written, so I support its publication.

I can only add a few comments to clarify and generate discussion on a couple of matters.

**Specific Comments**

Line 57: What is the reason that such a low correlation (r>0.32) is significant at 5% level? Very high variability?

The value of correlation required for statistical significance is determined by the sample size. With data for 39 winters available and neglecting interannual autocorrelation, there are 37 degrees for freedom in the calculation of the correlation coefficient. The weakest correlation that is significant at the 5% level is then slightly less than ±0.32 (e.g., https://www.real-
statistics.com/statistics-tables/pearsons-correlation-table/), assuming that the distribution of the data is not far from normal.

Lines 59-69: Although I agree that only "the first part of the reasoning is correct" regarding the analogy with the future climate, there is also observation-based research showing that snowfall and snow depth have already been increasing over some parts of Scandinavia and Eurasia (even if the reasons are not entirely clear). ,

I suggest to extend this paragraph and discuss these references too

Thanks for pointing out these references. I will discuss them in the revised manuscript, but I feel that they are easier to put in context after showing the projections from the EURO-CORDEX simulations. Therefore, I plan to leave this part of the Introduction nearly as is, except for replacing "earlier research suggest" (L64 in the original manuscript) with "climate model
projections suggest". The suggested references will be discussed in a new paragraph in the end of Section 6, planned to read approximately as follows. Note that the last sentence reflects one of your later comments!

A caveat in any model-based analysis is that climate changes in the real world may or may not follow the model projections. Interestingly, despite a decrease in winter mean and maximum snow depth in large parts of Europe since the 1950s (Fontrodona
Bach et al., 2018), Skaugen et al. (2012) found generally positive trends in winter maximum SWE above the 850 m altitude in southern Norway in the period 1931-2009. On a larger scale, Zhong et al. (2018) analysed observations of winter maximum snow depth in the Former Soviet Union, Mongolia and China, finding an average positive trend of 0.6 cm decade$^{-1}$ from 1966 through 2012. Increases in snow depth dominated especially north of 50°N, extending to milder regions than one would expect based on GCM projections for the future (Räisänen, 2008). Whether such differences reflect a problem in the models or have
resulted from multidecadal internal variability in the atmospheric circulation (Deser et al., 2012; Mankin and Diffenbaugh, 2015) is still an open question. If the atmospheric circulation turned out to be more sensitive to increasing greenhouse gas concentrations than current climate models indicate (as tentatively suggested by Scaife and Smith, 2018), some of the present conclusions might need to be modified.

55 March is chosen as a key month because of its maximum in SWE over most of the area, but March SWE is dependent on P and T of the previous winter months too.I n Figures 1,2,3 March SWE is compared to variability in NDJFM temperature and precipitation. However, in Figures 5, 9 and 12, it is not clear to me if the decomposition of variability into the three components is done only for March or for the entire winter. Perhaps this is clear from the mathematical theory presented, but a clarification and justification of this would be appreciated.

The diagnostic analysis integrates the effect of weather conditions from August until the month of interest (e.g., March), rather than using the data for this month alone. Furthermore, although the NDJFM season is used in some of the figures to provide and overview of the cold season weather conditions, it has no specific role in the calculation. To explain this better in the revised manuscript, I plan to revise the paragraph below Eq. (2) as follows:

Thus, the difference in SWE is decomposed to contributions from the differences in total precipitation ($\Delta P$), snowfall fraction ($\Delta F$) and the snow-on-ground fraction ($\Delta G$), plus a non-linear term that is typically much smaller than the first three right-hand-side terms in Eq. (2). As in Eq. (1), the time integrals in Eq. (2) start from August. The four right-hand-side (rhs) terms in Eq. (2) therefore integrate the effect of weather conditions from August until the month of interest (e.g., March), although the first month that matters in practice is the first month with non-zero mean snowfall. Thus, although the NDJFM season is used for characterizing the winter temperature and precipitation in some of the figures, the diagnostic analysis also uses data outside of this season.

Table 2: Might be my lack of understanding, but I do not know what the values in parentheses mean. What are the "individual terms"? How are they different from the four rhs terms in Eq. 2? It would be helpful to clarify this.

The values in the parentheses are the interannual standard deviation of each term, and its correlation with the actual SWE anomaly. To point out the connection to the earlier equations is an unambiguous way, the table caption will be modified as follows. Note that the earlier Eq. (4) will be Eq. (5) in the revised manuscript.

**Table 2.** Standard deviation (mm) of detrended March mean SWE anomalies in years 1982-2020 decomposed to its contributions from the four rhs terms in Eq. (2). The values in parentheses give the standard deviations of the individual terms ($s(\Delta SWE_i)$ in Eq. (5)) and their correlation with the SWE anomaly ($r(\Delta SWE_i, SWE)$ in Eq. (5)).

Lines 333-335. Regarding the positive correlation between SWE variability and temperature due to the westerly flow anomalies. Would the RCMs considered here, with boundary conditions from GCMs, reproduce any change in these anomalies which could have a strong effect in the future? Maybe just worth discussing this possibility.

I agree that it is prudent to leave the door open for the possibility that the weak circulation response in the models is incorrect.
To do this, I plan to add the following sentence in the very end of Section 6:

If the atmospheric circulation turned out to be more sensitive to increasing greenhouse gas concentrations than current climate models indicate (as tentatively suggested by Scaife and Smith, 2018), some of the present conclusions might need to be modified.

-Lines 354-356: Could this relate to the contrasting response of mean snow fall and extreme snowfall to warming as shown in https://doi.org/10.1038/nature13625 and O'Gorman https://doi.org/10.1007/s00382-015-2587-0 Räisänen

Yes, this may be part of the explanation, together with an overall decrease in the number of snowfall days and increased frequency of melt events. I plan to add a note on this in the second paragraph of Section 7 (the second sentence below):

This suggests that the snow conditions are becoming increasingly irregular, with an increasing number of virtually snow-free
winters but a smaller relative decrease in SWE in the most snow-rich winters than in an average winter. Apart from an increasing frequency of midwinter snowmelt events, this likely reflects an increase in relative snowfall variability as the number of days with snowfall decreases but the intensity of the largest snowfall events remains nearly unchanged (O' Gorman, 2014; Räisänen, 2016).

Given the choice of scenario RCP8.5, and the sensitivity of this type of research to crossing or not crossing the freezing threshold (snow or no snow), it would be good to raise a point in the conclusions whether how different might the results under another scenario. Or to call for future work on the analysis of multiple scenarios.

If the greenhouse gas emissions were smaller than those in RCP8.5, all the climate changes, including the rate at which SWE
decreases, would mostly likely be smaller. It would also take longer for the changes to emerge clearly from the background of natural variability. On the other hand, the projections for the mid-century period 2020/21-2058/59 are qualitatively similar to those for 2059/60-2097/98, although the radiative forcing is much weaker. This suggests that the basic, qualitative nature of the changes should not be very sensitive to the choice of the scenario, although their magnitude is. Considering this comment, I plan to extend the second last paragraph in the Conclusions section as follows:

… These considerations qualitatively explain both the geographical contrasts in the drivers of the present-day SWE variability and the shift towards increasingly snow-on-ground and snowfall fraction dominated SWE variability in a warmer future climate. Under a scenario with smaller greenhouse gas emissions, this shift as well as the changes in mean SWE would most likely proceed more slowly than the present results for RCP8.5 indicate, and it would take longer for them to rise above the
background of natural variability. However, the qualitative similarity between the multi-RCM mean projections for 2059/60-2097/98 and the midway period 2020/21-2058/59 (Figs. 9, 10, 12 and 13) suggests that the basic characteristics of these changes should be largely insensitive to the magnitude of the radiative forcing.

Note that, in response to the comments of Reviewer 1, I will add a new Fig. 8, and the figure numbers thereafter are changed
accordingly.

**Technical corrections:**

Line 57 in caption should be: (c) NDJFM mean precipitation (not temperature)

Will be corrected.

Lines 150-154: Please add also "Equation" to "1" and "2", to clarify.-

Sorry for being unclear with the notations. "1" and "2" here referred to the subscripts introduced just above Eq. (1), not to the numbers of equations. The planned clarification is as follows:

In this study, the decomposition (2) is applied in two different ways:

1.  When studying interannual variations in SWE, $X_1$ as defined above Eq. (2) represent the mean values for a 39-winter period (1981/82 to 2019/20, 2020/21 to 2058/59 or 2059/60 to 2097/98) and $X_2$ the values for an individual winter.
2.  When studying long-term changes in SWE, $X_1$ represent the mean values for winters 1981/82 to 2019/20, and $X_2$ those for either 2020/21 to 2058/59 or 2059/60 to 2097/98.

Thanks for noticing this. Will be corrected.

---

## Author Response (AR2)

Dear Dr. Niwano,

Thanks for checking my revised manuscript and for your three suggestions for improved wording. I have accepted all these suggestions.
* * *
L. 89 ~ 90: "the description of snowmelt is important for the simulation of both the interannual variability and future trends of snow amount.": The intention of "the description of snowmelt" is unclear. I suggest rephrasing the part to something like "it is imperative to calculate snowmelt accurately for the realistic simulations of both the interannual variability and future trends of snow amount.".

L. 139: Suggest rephrasing "since precipitation measurements in Finland only record the total precipitation." -> "since precipitation measurements in Finland only record the total amounts.".

L. 393 ~ 394: Suggest rephrasing "~ anomalies in ERA5-Land would suggest." -> "~ anomalies that ERA5-Land indicates.". Using "suggest" sounds weak in this case. I think a stronger word like "indicate" is better here.
* * *
The line numbers for these changes in the revised *pdf file* are the same as above (89-90, 139 and 393-394). The corresponding line numbers in the *text file* (.docx), in which the figure captions and the tables have been moved to the end, are 80-81, 125 and 341-342, respectively.

Sincerely,

Jouni Räisänen